# Linking individual differences in human primary visual cortex to contrast sensitivity around the visual field

Marc M. Himmelberg ⬤ [1,2 ✉], Jonathan Winawer ⬤ [1,2,3] & Marisa Carrasco ⬤ [1,2,3]

A central question in neuroscience is how the organization of cortical maps relates to perception, for which human primary visual cortex (V1) is an ideal model system. V1 non-uniformly samples the retinal image, with greater cortical magnification (surface area per degree of visual field) at the fovea than periphery and at the horizontal than vertical meridian. Moreover, the size and cortical magnification of V1 varies greatly across individuals. Here, we used fMRI and psychophysics in the same observers to quantify individual differences in V1 cortical magnification and contrast sensitivity at the four polar angle meridians. Across observers, the overall size of V1 and localized cortical magnification positively correlated with contrast sensitivity. Moreover, greater cortical magnification and higher contrast sensitivity at the horizontal than the vertical meridian were strongly correlated. These data reveal a link between cortical anatomy and visual perception at the level of individual observer and stimulus location.

---

[1] Department of Psychology, New York University, New York, NY 10003, USA. [2] Center for Neural Science, New York University, New York, NY 10003, USA. [3] These authors contributed equally: Jonathan Winawer, Marisa Carrasco. ✉email: marc.himmelberg@nyu.edu

Human primary visual cortex (V1) is an ideal model system for investigating the link between cortical anatomy and visual perception. The size and organization of human V1 varies across individuals. V1 surface area varies more than two-fold across individuals[1–6]. V1 surface area also varies within individuals. Specifically, the amount of V1 tissue dedicated to processing a fixed spatial extent on the retina (cortical magnification) changes sharply as a function of eccentricity and polar angle[7–11]. Moreover, within- and between-individual variation interacts: even after accounting for differences in overall V1 size between observers, the cortical magnification functions for eccentricity and polar angle vary across individuals[3,10].

Variation in cortical magnification has important consequences for visual perception. In human and non-human primates, visual performance is typically best near the fovea and decreases with increasing eccentricity[12–18]. This is reflected in the cortical magnification function, in which V1 surface area is greatest at the fovea and decreases with increasing eccentricity[2,7,9,11,19–21]. Visual performance for many tasks (including contrast sensitivity) also changes as a function of polar angle; it is better along the horizontal than vertical meridian (horizontal-vertical anisotropy, HVA), and along the lower than upper vertical meridian (vertical meridian asymmetry, VMA)[12,22–27]. These perceptual polar angle asymmetries have been linked to similar asymmetries in V1 cortical magnification: in humans and macaque monkey, more local V1 surface area is dedicated to processing the horizontal than vertical meridian (i.e., a cortical HVA) and the lower than upper vertical meridian (i.e., a cortical VMA)[8–10,28]. Furthermore, polar angle asymmetries in cortical magnification are at least in part heritable—the magnitudes of the asymmetries are more similar for monozygotic than dizygotic twins[10]. Individual variation in V1 cortical magnification (both the total size of V1 and how its surface area is distributed across the visual field map) may account for individual differences in visual perception.

Contrast is the currency of the visual system, driving neural and perceptual responses for most, if not all, visual tasks. V1 neurons are highly sensitive to contrast[29–34], with their firing rates reaching half of their maximal response ($c_{50}$) at contrasts as low as ~4%[29,35]. Contrast sensitivity and cortical magnification co-vary as a function of eccentricity (i.e., contrast sensitivity and cortical magnification decrease with increasing eccentricity)[36]. A possible linking hypothesis connecting these two observations is that contrast sensitivity is determined by the number of activated V1 neurons[36]. As the cytoarchitecture of V1 is approximately uniform[37,38], portions of V1 with substantially more dedicated cortical surface area per square degree of visual field should also have more neurons[39], and contrast sensitivity at some location in the visual field should increase in proportion to the amount of cortical surface area dedicated to encoding that location, thereby linking cortical magnification with contrast sensitivity.

This linking hypothesis was originally proposed by Virsu and Rovamo[36] to explain changes in contrast sensitivity as a function of eccentricity. Here, we extend this logic to individual variation in V1 size, and to local measurements of V1 surface area taken as a function of polar angle, across observers.

To assess the link between contrast sensitivity and cortical magnification, 29 observers completed a psychophysical orientation discrimination task to measure contrast sensitivity at the four polar angle meridians in the visual field (left and right horizontal, upper vertical, and lower vertical). In the same observers, we then used an fMRI retinotopic mapping experiment to delineate V1 maps. We calculated the V1 surface area (out to 8° eccentricity) and the surface area of "wedge-ROIs", defined as the portions of V1 dedicated to processing ±15° regions of the visual field centered along the same four meridians as in the contrast sensitivity measurements. As the wedge-ROIs always

represent the same size region of visual space (±15° of polar angle, 1–8° of eccentricity), any differences in V1 surface area measurements derived from the wedge-ROIs can be considered to index differences in cortical magnification.

To preview our results, our data show that: First, there was a positive correlation between contrast sensitivity (averaged across polar angle location) and the overall size of V1; observers with greater contrast sensitivity tended to have a larger V1, and vice versa. Second, there was a positive correlation between local contrast sensitivity and local V1 surface area measurements taken from the polar angle meridians; observers with greater contrast sensitivity measurements at some polar angle locations tended to have more local V1 surface area dedicated to encoding that location. Third, a stronger HVA for contrast sensitivity positively correlated with the corresponding asymmetry in the distribution of local V1 surface area, however, this was not the case for the VMA. Together, these findings provide support for the hypotheses linking contrast sensitivity to cortical magnification[36]. Further, they reveal that perceptual polar angle asymmetries are grounded in the asymmetric distribution of cortical tissue of V1, and more broadly, provide a link between visual perception and the idiosyncratic organization of V1.

## Results

**The size of V1 varies substantially across observers.** First, we assessed the distribution of V1 surface area across 29 observers. Here, we report V1 surface area per hemisphere, from 0° to 8° eccentricity, which comprises almost half of V1. We limit the eccentricity range to 8° because the functional data are less reliable near the edge of the retinotopic mapping stimulus (12.4°)[9,40] and to match the eccentricity extent of the wedge-ROIs used in the cortical magnification analysis. Consistent with previous reports, the surface area of V1 varied ~twofold (specifically by 110%, the largest V1 being 1776 mm$^2$ and the smallest being 832 mm$^2$) (Fig. 1a). The variability in V1 surface area is substantial, especially when compared to the total surface area of the cortex (Fig. 1b), which varied by 50% (the largest hemisphere of the cortex being 0.12 m$^2$ and the smallest being 0.08 m$^2$). Figure 1d shows visualizations of polar angle and eccentricity maps from the left V1 for the largest and smallest V1s. Both had clear, full representations of the right visual hemifield, indicating that the visual field maps were clear and complete, and there were no errors in delineating their boundaries. Instead, the retinotopic representations are simply compressed for the observer with a smaller V1. These large individual differences are not a result of sex differences. There was little difference in the size of V1 between females and males, regardless of whether we normalize the size of V1 to total cortical surface area ($p > 0.1$ for both comparisons, unpaired two-tailed $t$ tests).

Although V1 surface area varies across individuals, the surface areas of left and right V1 are relatively similar within individuals[1,2,4]. Here, too, we found that the surface area of left and right V1 were highly correlated ($r = 0.67$, $p < 0.001$, Fig. 1c).

**Replicating group-level polar angle asymmetries for contrast sensitivity and V1 surface area.** We tested whether the expected polar angle asymmetries existed in the psychophysical and cortical data at the group level. First, to calculate contrast sensitivity along the horizontal meridian, we averaged contrast sensitivity measurements from the left and right horizontal meridians. Similarly, to calculate contrast sensitivity along the vertical meridian, we averaged together contrast sensitivity measurements from the upper and lower vertical meridians. To calculate the amount of V1 surface area dedicated to processing the horizontal meridian, surface area measurements from the wedge-ROIs

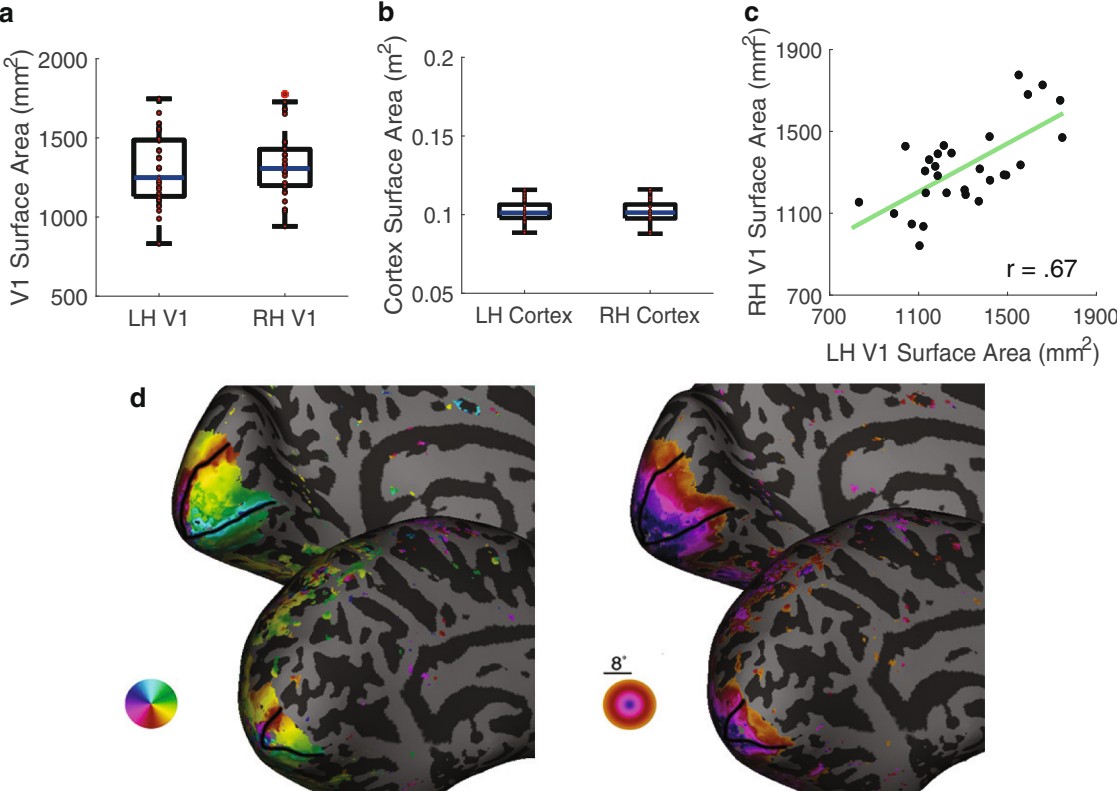

**Fig. 1 Variability of human primary visual cortex. a** Surface area (mm$^2$) for the left and right hemispheres of V1 ($n = 29$) and **b** the total surface area for left and right hemispheres of the cortex ($n = 29$). The $y$-axes are matched so that the scaling of the values in **b** are 100× greater than those in **a**. Individual data are plotted in red. The horizontal line represents the median. Top and bottom bounds of each box represent the 75th and 25th percentiles, respectively. The whiskers extend to the minima and maxima data points not considered outliers. **c** The surface area of the left and right hemispheres of V1 are strongly correlated within individuals (two tailed Pearson's correlation, $r_\rho = 0.67$, $p < 0.001$). **d** Polar angle and eccentricity maps on the inflated left hemisphere for the individuals with the largest and smallest V1s. The border of V1 is defined by the black lines and data are shown out to 8° of eccentricity. Source data for **a**, **b** and **c** are provided as a Source Data file.

centered on the left and right horizontal meridians were summed. Similarly, to calculate the amount of surface area dedicated to processing the vertical meridian, the V1 surface area measurements from the wedge-ROIs centered on the upper and lower vertical meridians were summed. See *Methods* section *Defining wedge-ROIs* for further details.

As expected, both the HVA and VMA emerged at the group-level in the contrast sensitivity data. Contrast sensitivity was significantly greater at the horizontal than the vertical meridian (HVA) ($t(28) = 15.88$, $p < 0.001$, $d = 2.95$; paired samples $t$ test; Fig. 2a). Similarly, contrast sensitivity was greater at the lower than the upper vertical meridian (VMA) ($t(28) = 4.18$, $p < 0.001$, $d = 0.78$; Fig. 2a). Both the HVA and VMA were well-correlated across subsampled blocks of the behavioral data from each observer, indicating that the contrast sensitivity measurements were reliable within each observer and supporting the use of contrast sensitivity measures as reflecting genuine individual differences (Supplementary Fig. 1).

Likewise, we confirmed the HVA and VMA at the group level in the V1 surface area data. There was significantly more V1 surface area dedicated to the horizontal than vertical meridian (HVA) ($t(28) = 16.40$, $p < 0.001$, $d = 3.05$; Fig. 2b). Similarly, there was more V1 surface area dedicated to the lower than upper vertical meridian (VMA) ($t(28) = 3.30$, $p = 0.002$, $d = 0.61$; Fig. 2b). Neither the effect of sex nor its interaction with location (HVA or VMA) for contrast sensitivity or surface area was significant ($p > 0.1$ for main effect of sex and interactions; 2 × 2-way (2 locations × 2 sexes) repeated measures ANOVAs).

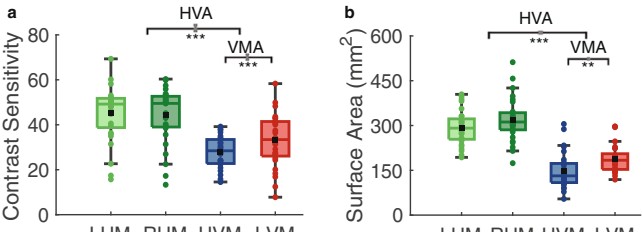

**Fig. 2 Group-level polar angle asymmetries for contrast sensitivity and V1 surface area. a** Group-level contrast sensitivity measurements from the four polar angle meridians (paired samples $t$-tests, two sided) and **b** group-level V1 surface area measurements from wedge-ROIs (±15°) centered on the four polar angle meridians (paired samples $t$-tests, two sided) ($n = 29$). Colored data points indicate individual measurements. The black datapoint represents the group average and the colored horizontal line represents the group median. Gray error bars on the horizontal brackets represent ±1 standard error of the difference. Top and bottom bounds of each box represent the 75th and 25th percentiles, respectively. The whiskers extend to the minima and maxima data points not considered outliers. \*\*$p < 0.01$, \*\*\*$p < 0.001$. Source data are provided as a Source Data file.

Additionally, we identified slightly more V1 surface area dedicated to the right than the left horizontal meridian ($p = 0.028$, $d = 0.43$). In a supplementary analysis we tested this left-right horizontal meridian asymmetry using an extended dataset

(n = 54; the 29 observers here and 25 additional observers for whom we have retinotopy measurements but no psychophysics data) and found the slight bias for more surface area along the right than left horizontal meridian was diminished and became only marginally significant (p = 0.065), d = 0.26; Supplementary Fig. 2).

**Overall V1 surface area predicts contrast sensitivity.** After confirming the polar angle asymmetries at the group level, we measured the relation between contrast sensitivity and V1 surface area at different scales, across individual observers. The relation between these two variables is non-linear; an increase in V1 surface area does not necessarily correspond to a proportional increase in contrast sensitivity. However, the relation between these two variables is hypothesized to be monotonic and positive; as surface area increases, so will contrast sensitivity. Thus, the following analyses were conducted using one-tailed Spearman's rho; $r_\rho$[41].

First, we asked whether contrast sensitivity averaged across the four polar angle locations correlated with overall surface area (i.e., the size) of V1 (summed across hemispheres and restricted to 0–8° of eccentricity). Averaged contrast sensitivity across location was positively correlated with V1 surface area ($r_\rho = 0.47$, p = 0.005; Fig. 3). The surface area of a cortical region depends on the cortical depth used to define the surface. The superficial surface is an overall larger surface area than a surface defined at a deeper layer. Moreover, the superficial layer has greater surface area for gyri and the deeper surfaces have greater surface area for sulci. To reduce these biases, we defined surface area on the midgray surface, half-way between the white matter and the pial surface. We also measured all effects relative to the white matter surface and the pial surface and found a consistent pattern of results; significant correlations were also found when V1 surface area was calculated using the pial (Supplementary Fig. 3a) or white matter surfaces (Supplementary Fig. 3b), rather than the midgray (see *Methods: Midgray, pial, and white matter surfaces*). Overall, observers with a larger V1 tended to have greater contrast sensitivity, whereas those with a smaller V1 tended to have relatively lower contrast sensitivity.

When we computed the relation between contrast sensitivity and V1 surface area, we did not normalize V1 surface area by the total cortical surface per observer. In rodents, animals with larger brains have larger neurons, so the number of neurons is approximately constant despite differences in brain size[42]. Were this also the case for the human brain, it would be appropriate to normalize each observer's V1 surface area by their total cortical surface area, as in this case surface area would be a good proxy for neural count. When we normalize by total cortical surface area and then correlate with contrast sensitivity, a positive correlation between the variables remains ($r_\rho = 0.33$, p = 0.038; Fig. S4). Further, average contrast sensitivity did not correlate with overall cortical surface area (p = 0.106), indicating that these correlations are specific to V1.

**Local V1 surface area measurements predict contrast sensitivity at the polar angle meridians.** Next, we assessed the relation between contrast sensitivity and V1 surface area at a finer granularity. Across observers, we asked whether contrast sensitivity at each polar angle meridian correlated with V1 surface area measurements taken from the spatially corresponding wedge-ROI. Each wedge-ROI was ±15° in width, extended from 1° to 8° eccentricity, and was centered along a polar angle meridian corresponding to the contrast sensitivity measurements. Spearman's rho was used to assess the relation between contrast sensitivity measurements taken at each of the left and right horizontal, upper, and lower vertical meridians and the surface area of the ±15° wedge-ROI centered on the corresponding meridian.

Across polar angle locations and observers, contrast sensitivity measurements were strongly correlated with the corresponding V1 surface area ($r_\rho = 0.60$; Fig. 4a). Note that this correlation relies on the variability across two factors: polar angle and individual observer. Thus the data points were not independent, with a data point from each observer per polar angle location. To assess whether each of these factors significantly contributed to the correlation, we computed two null distributions (Supplementary Fig. 5.). One removed variability across observers and the other removed variability across polar angle. Each distribution was generated by bootstrapping 10,000 Spearman's rho correlations to the contrast sensitivity and localized surface area measurements. The first null distribution removed variability across observers: on each iteration we shuffled the assignment of the four contrast sensitivities and four local surface area values across observers, while maintaining the tie of a given quadruple of contrast sensitivity and local surface area measurements at each location. The second null distribution removed variability across polar angle: on each iteration we shuffled the assignment of the four contrast sensitivities and four local surface area values across locations, while maintaining the tie of a given quadruple of contrast sensitivity and local surface area measurements to an observer. We then calculated the $r_\rho$ values at the 95th percentile ($x_{0.95}$)[43] of these bootstrapped $r_\rho$ distributions.

For the null distributions, the $r_\rho$ values at the 95th percentile ($x_{0.95}$) were 0.56 (first distribution, removing the effect of individual observer) and 0.24 (second distribution, removing the effect of polar angle). Both of these values are less than the $r_\rho$ value of 0.60, obtained from the unshuffled distribution, thereby demonstrating a significant contribution of both polar angle and individual observer to the correlation. A similar correlation between contrast sensitivity and local surface area was found using measurements on the pial surface (Supplementary Fig. 6a) and white matter surface (Supplementary Fig. 6b). Thus, local V1 surface area (or equivalently, cortical magnification) predicted contrast sensitivity measurements taken from different polar angle locations.

To visualize the contribution of variability across polar angle and individual observer, we factored out between-observer variability or within-observer variability from the data. We did this by subtracting the contrast sensitivity/surface area value averaged across the four polar angle locations separately for each observer (Fig. 4b) or by subtracting the contrast sensitivity/surface

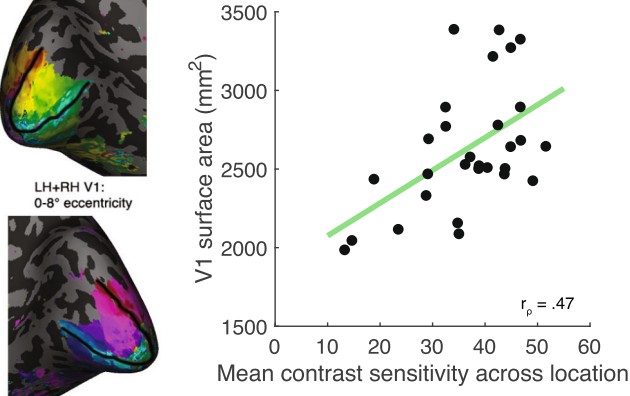

**Fig. 3 Individual differences in V1 size and contrast sensitivity.** Average contrast sensitivity across polar angle locations correlates with V1 midgray surface area (Spearman's correlation, one-tailed, $r_\rho = 0.47$, p = 0.005, n = 29). For each observer, contrast sensitivity is defined as contrast sensitivity averaged across the four polar angle locations and V1 size is defined as the summed cortical surface area of left and right V1 within 0–8° eccentricity, as shown in the inset panel. Source data are provided as a Source Data file.

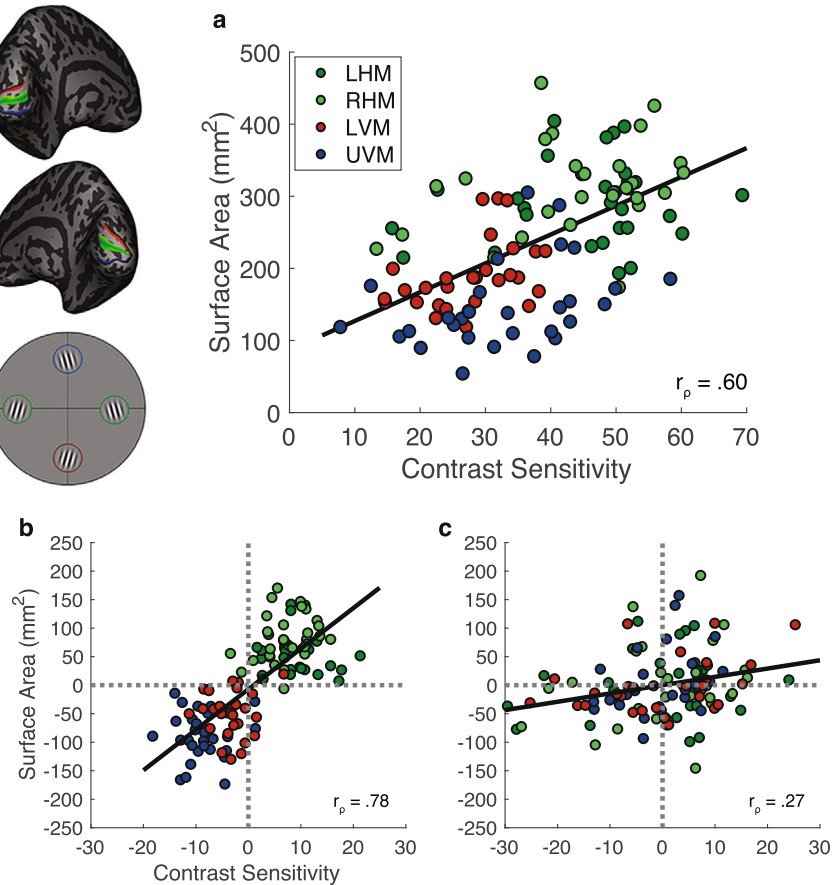

**Fig. 4 Correlations between spatially localized contrast sensitivity and V1 surface area measurements. a** Contrast sensitivity correlates with local V1 surface area (i.e., cortical magnification) measurements taken from the corresponding meridians ($n = 29$). We correlate contrast sensitivity and V1 surface area of ±15° wedges centered on the respective polar angle meridians (Spearman's correlation, one-tailed, $r_\rho = 0.60$, $x_{0.95} = 0.56$ and 0.24). **b** and **c** show correlations assessing the contribution of within-observer variability (polar angle meridian) and between-observer variability (individual differences) to the relation between local contrast sensitivity and local V1 surface area. In **b** the between-observer variability is removed from the correlation to show the effect of meridian (Spearman's correlation, one-tailed, $r_\rho = 0.78$), whereas in **c** the within-observer variability is removed from the correlation to show the effect of individual observers (Spearman's correlation, one-tailed, $r_\rho = 0.27$). Data are color-coded to reflect the meridian from which they come from; green data come from the left and right horizontal meridian, red data come from the lower vertical meridian, and blue data come from the upper vertical meridian. Source data for **a**, **b** and **c** are provided as a Source Data file.

area value averaged across the 29 observers separately for each of the four polar angles (Fig. 4c). In both cases, a positive correlation remained, supporting the statistical comparisons to the null distributions described above. In particular, these results show a robust effect of polar angle on the data ($r_\rho = 0.78$; Fig. 4b) supporting the statistical comparison, and a modest effect of individual observer variability on the data ($r_\rho = 0.27$; Fig. 4c).

**Quantifying the relation between the strengths of the behavioral and cortical HVA and VMA**. Next, we calculated a summary metric to describe the strength of the HVA and VMA for contrast sensitivity and local surface area. We then assessed the relation between the strength of behavioral and cortical HVA and VMA across observers.

For each observer, we calculated an asymmetry index for the HVA. The HVA index was calculated as the difference in contrast sensitivity or local V1 surface area between the horizontal and vertical meridian, divided by the mean of the two, multiplied by 100.

$$\text{HVA} = \frac{(\text{horizontal} - \text{vertical})}{\text{mean}(\text{horizontal, vertical})} \times 100 \quad (1)$$

An HVA index of 0 indicates no difference in contrast sensitivity or surface area between the horizontal and vertical

meridian. As the HVA index increases, the asymmetry increases, with greater contrast sensitivity, or surface area, at the horizontal than vertical meridian.

Correspondingly, the VMA index was calculated as the difference in contrast sensitivity or local V1 surface area between the lower and upper vertical meridian, divided by the mean of the two, multiplied by 100.

$$\text{VMA} = \frac{(\text{lower vertical} - \text{upper vertical})}{\text{mean}(\text{lower vertical, upper vertical})} \times 100 \quad (2)$$

A VMA index of 0 indicates no difference in contrast sensitivity or surface area between the lower and upper vertical meridian. As the VMA index increases, the asymmetry increases, with greater contrast sensitivity, or surface area, at the lower than upper vertical meridian.

Importantly, these HVA and VMA strengths reflect differences in contrast sensitivity, or V1 surface area, between locations, after dividing out the mean contrast sensitivity or local V1 surface area, per observer. Therefore, one could have low contrast sensitivity measurements, but a strong HVA.

Spearman's rho was used to assess the relation between the HVA index for contrast sensitivity and local V1 surface area across observers. There was a significant, positive correlation

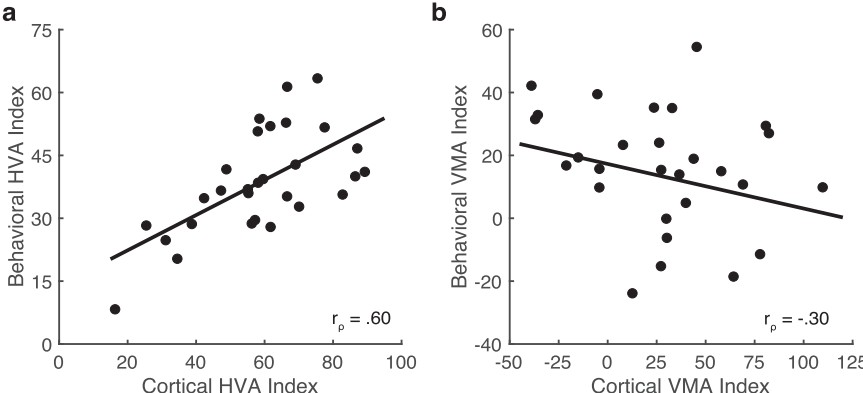

**Fig. 5 Correlations between the strengths of the behavioral and cortical HVA and VMA. a** The behavioral and cortical horizontal-vertical anisotropy (HVA) correlate (Spearman's correlation, one-tailed, $r_\rho = 0.60$, $p < 0.001$), **b** however the behavioral and cortical vertical meridian asymmetry (VMA) do not (Spearman's correlation, one-tailed, $r_\rho = -0.30$, $p = 0.942$) ($n = 29$). Source data for **a** and **b** are provided as a Source Data file.

between the two measurements ($r_\rho = 0.60$, $p < 0.001$; Fig. 5a). A similar correlation was found when the wedge-ROIs were used to calculate V1 surface area measurements on the pial (Supplementary Fig. 7a) and white matter surfaces (Supplementary Fig. 7c). Thus, observers with a stronger asymmetry in contrast sensitivity measurements between the horizontal and vertical meridians had a stronger asymmetry in dedicated V1 surface area between the horizontal and vertical meridians.

Following this, Spearman's rho was used to test the relation between the VMA index for contrast sensitivity and local V1 surface area. There was a non-significant correlation between the two measurements ($r_\rho = -0.30$, $p = 0.942$; Fig. 5b). Thus, the individual asymmetry for contrast sensitivity between the lower and upper vertical meridian was not associated with the individual amount of V1 surface area dedicated to the lower and upper vertical meridian. There was also no systematic relation when the local V1 surface area measurements were made on the pial surface (Supplementary Fig. 7b) and white matter surface (Supplementary Fig. 7d). In a supplemental analysis, we found no significant correlation between the behavioral HVA and VMA (Supplementary Fig. 8a) and only a marginal correlation between the cortical HVA and VMA (Supplementary Fig. 8b).

## Discussion

We quantified the relation between contrast sensitivity and V1 surface area at a global scale (i.e., the surface area of V1 itself) and a local scale (i.e., the local surface area of V1 processing the polar angle meridians) across 29 observers. We confirmed group-level polar angle asymmetries in the contrast sensitivity and V1 surface area data. We then quantified individual differences across observers, leading to three major findings. First, contrast sensitivity averaged across the four polar angle locations positively correlated with the size of V1. Second, contrast sensitivity measurements taken at the four polar angle locations positively correlated with localized V1 surface area measurements taken from the spatially corresponding polar angle meridian in the visual field. Third, the extent of the HVA for contrast sensitivity was correlated with the extent of the HVA for V1 surface area, whereas the VMA was not.

**Group-level reproduction of polar angle asymmetries**. The data showed clear group-level polar angle asymmetries for contrast sensitivity and V1 surface area. Contrast sensitivity was roughly 50% higher for the horizontal than vertical meridian and 20% higher for the lower than upper vertical meridian. Likewise, the V1 surface area measurements also showed group-level polar

angle asymmetries; V1 surface area was around 60% greater along the horizontal than vertical meridian and around 25% greater along the lower than upper vertical meridian, consistent with data from our previous study[9], the Human Connectome Dataset[10] and other work[28,44]. Polar angle asymmetries have also been found in the surface areas of non-human primate V1[8], in the amplitude of the BOLD response in human V1[44,45], and in spatial frequency preference in human[46]. The existence of these polar angle asymmetries in multiple, large datasets for both the behavioral and cortical data indicates that they are robust at the group-level, and speaks to the high level of reproducibility reported for fMRI-derived retinotopic maps[2,7,9,11,19–21].

The present group-level data showed a small bias towards more V1 surface area dedicated to the right than left horizontal meridian. It might be that a left-right horizontal meridian asymmetry relates to visual tasks in which an advantage along the right horizontal meridian exists, such as crowding[47–49] and letter recognition[50,51]. A larger left than right hemisphere of V1 has been previously reported[4] (but see refs. [2,28]). However, none of these studies examined the surface area of the horizontal meridian specifically.

**Variation in the size of V1 surface area across observers**. The surface area of V1 varied substantially across observers, whereas within observers, the surface area of V1 in the left and right hemispheres was relatively consistent, in line with previous studies[1,2,4,5,8]. V1 size is only weakly correlated with overall cortical surface area, which is less variable in size, as shown here and in prior reports[2,52]. Neither does V1 size differ between males and females after normalization to total cortical surface area, again shown here and elsewhere[2,10]. Why is there so much variability in the size of V1? One hypothesis is that variation in the size of V1 depends on the amount of detail encoded in earlier stages of the visual system: cone density varies by about threefold across observers[53] and the size of the LGN and optic tract also vary substantially[1,54]. Indeed, the size of V1 correlates with the size of the lateral geniculate nucleus and the optic tract, suggesting that these components of the early visual system, all of which are important for visual perception, develop interdependently[1,54]. This finding lends credence to the possibility that the size of V1 is important for perceptual tasks.

**Greater contrast sensitivity is a perceptual consequence of greater V1 surface area**. Here, we have shown that contrast sensitivity measurements derived from an orientation discrimination task positively correlate with V1 surface area;

observers with greater contrast sensitivity tend to have a larger V1 and those with lower contrast sensitivity tend to have a smaller V1. This relation holds irrespective of the cortical depth used to compute surface area (gray/pial boundary, gray/white boundary, or half-way between them). V1 surface area has been shown to correlate with a few measurements of visual performance, such as perceptual acuity thresholds[55,56], measurements of subjective object size[52], and orientation discrimination thresholds[57], but not with contrast discrimination thresholds[57]. Such thresholds are different from those measured here; they depend upon the range of the contrast response function being measured[58,59], but not upon stimulus orientation. Nonetheless, performance on most visual tasks has not been compared to V1 size and there is not yet a computational account that would enable one to predict to what extent, if any, performance on different tasks would be affected by V1 size.

We focused on the relation between performance on one visual measure—contrast sensitivity—and the size of the one cortical map—V1. Although V1 size has been linked to a few perceptual measures[52,55–57,60–62], it is likely that there will be other measures for which performance is better explained by the size of other visual maps. Such an outcome is possible because the sizes of different maps are at least partially independent[2,4]. For example, an observer might have a large V1 and high contrast sensitivity, but a small hV4 and poor performance on visual crowding tasks[63].

An open question is whether and how the variation in the size of V1 relates to neural circuitry. Smaller V1s have a full, but relatively compressed representation of the visual field. Does this compression represent fewer overall neurons or is neural count similar across individuals and instead this compression represents increased neural density? The fact that performance on some tasks correlates with V1 surface area[52,55–57] suggests that a smaller V1 likely has fewer overall neurons, but the histological measures to directly assess this do not yet exist. Differences in V1 neural counts among individuals and across polar angle raise the question of how the neural code varies across individuals and visual field location. One interesting observation is that the size of V1 is inversely correlated with the size of its population receptive fields (pRFs), suggesting that a larger V1 enables finer sampling of visual space[64].

**Local contrast sensitivity is linked to local V1 cortical magnification around the visual field.** Virsu and Rovamo[36] hypothesized that the mechanism underlying contrast sensitivity is a central integrator that pools the activity of V1 neurons; contrast sensitivity should increase in proportion to local cortical surface area (i.e., cortical magnification) and thus the number of neurons activated by a visual stimulus. This hypothesis was derived from group-level behavioral measurements taken as a function of eccentricity. We have tested whether this hypothesis holds for individual, localized V1 surface area measurements taken as a function of polar angle, as well as for individual measurements of the size of V1. We found that observers with more local cortical surface area dedicated to processing some polar angle location had greater contrast sensitivity at the corresponding location, and that individuals with larger V1 had overall higher contrast sensitivity. Therefore, our data support the hypothesis that contrast sensitivity varies as a function of the number of stimulated visual neurons.

Perceptual measurements have been related to the surface area of entire visual maps[52,55–57]. Advances in computational neuroimaging provide the tools to precisely delineate visual maps and assess their internal layout[65], enabling the assessment of the relation between cortical anatomy and performance as a function

of location in the visual field. Indeed, three studies have related individual differences in localized measurements of cortical magnification to perceptual outcomes. Local V1 cortical magnification positively correlates with visual acuity measured as a function of eccentricity[3], position discrimination ability at different angular locations[55], and subjective object size for different visual field quadrants[66].

**Performance field asymmetries.** Perceptual polar angle asymmetries (i.e., performance fields) have been established across a broad range of visual tasks. The HVA and VMA are found in tasks involving contrast sensitivity[12,22,23,67–73], perceived contrast[74], spatial resolution[24,75,76], crowding[49], temporal information accrual[77], illusory motion perception[25], and visual short term memory[78]. Further, these polar angle asymmetries are pervasive across a range of conditions; they exist across luminance levels[22], binocular and monocular stimulation[22,24], different stimulus orientations[22,68,73] and sizes[12], eccentricities and spatial frequencies[12,22–24,68], number of distractors[22,27,77], and covert attentional conditions[22,23,27].

We found no relation between the behavioral HVA and VMA, consistent with previous reports[12,24], and only a weak, non-significant relation between the cortical HVA and VMA. These results suggest that the HVA and VMA are independent of each other at the level of perception and at the level of the cortex. It is likely that these two behavioral asymmetries develop with age independently[79] and may have separate neural substrates. Cortical magnification changes as a function of eccentricity in a similar fashion for children and adults[80]; However, perceptual polar angle asymmetries vary between children and adults[81] and how cortical magnification changes as a function of polar angle in children still needs to be determined. Furthermore, the HVA, but not VMA, exists in photoreceptor cone density[82,83], whereas both the HVA and VMA exist in retinal midget ganglion cell density[84,85]. A computational model has shown that the HVA and VMA for contrast sensitivity cannot be fully explained by these retinal factors. Asymmetries in optics and cone sampling only accounted for a small fraction of contrast sensitivity asymmetries[86]. Including midget retinal ganglion cells in the model explained a larger fraction of the contrast sensitivity asymmetries but did not account for the extent of the asymmetries reported in human visual behavior[87]. Thus, these retinal asymmetries did not account for perceptual polar angle asymmetries at the group level. It is unlikely that retinal asymmetries could account for individual polar angle asymmetries.

**Individual differences in perceptual polar angle asymmetries are rooted in individual variation in cortical anatomy.** We have shown that the HVA for contrast sensitivity can be predicted from the cortical HVA in individual observers. Thus, the perceptual asymmetry between the horizontal and vertical meridian for contrast sensitivity is strongly reflected by the relative distribution of V1 surface area between the horizontal and vertical meridian. The findings that the behavioral and cortical HVA can be linked across individual observers, and is stronger in the visual cortex than the retina[10,87], suggest that this perceptual asymmetry can be predominantly explained by the asymmetric distribution of cortical surface (and thus neurons) in V1.

Here, we found group-level VMA measurements for contrast sensitivity and surface area, and it has been shown that group-level VMA measurements for spatial acuity thresholds and V1 surface area correlate[10]. However, we did not find a relation between the VMA for contrast sensitivity and the cortical VMA at the level of individual observers. Why might this be? One possibility is statistical power. The VMA was computed using half

the amount of data as the HVA (i.e., the upper vs lower vertical meridian, rather than the left and right horizontal combined vs the upper and lower vertical combined) and the strength of the VMA was about half the size of the HVA. Another possibility is measurement constraints; the vertical meridian lies at the extreme range of the polar angle distribution within a hemisphere. Therefore, the blurring of neural responses due to the fMRI measure will skew pRF centers away from the vertical meridian representation. Many studies have noted a lack of representation of the visual field close to the vertical meridian in V1 and other visual field maps, presumably for this reason[40,88–90]. This limitation introduces some noise into estimates of the size of the cortical representation of the vertical meridian, compounding with the reduced SNR due to an overall smaller representation of the vertical meridian. Alternatively, individual differences in the behavioral VMA might be explained, at least in part, by factors other than the total computational resources (e.g., surface area) afforded by V1. For example, individual differences in the VMA might be dependent upon the tuning properties of V1 neurons or by the efficiency of how V1 outputs are readout by neurons in downstream visual maps.

**Extending the link between brain and behavior.** Here, we have linked V1 cortical magnification to contrast sensitivity for a particular stimulus configuration (a 3° vertically oriented Gabor with a spatial frequency of 4 cpd centered at 4.5° eccentricity on a uniform gray background). Would these results generalize to other orientations, spatial frequencies, and stimulus sizes? Perceptual polar angle asymmetries are robust across modulations of stimulus content for which V1 neurons are tuned. They persist across different stimulus orientations[22,68,73], sizes[12], eccentricities and spatial frequencies[12,22–24,68], and in the presence of distractors[22,27,77]. Likewise, cortical polar angle asymmetries have been reproduced across several independent datasets that differ in their experimental design, including differences in the pRF stimulus carrier image[9,10,28]. The cortical asymmetries are robust to experimental differences because they rely on polar angle pRF measurements that have shown to be highly reproducible across retinotopy experiments[9]. As these behavioral and cortical asymmetries are preserved across an array of stimulus conditions, we predict that the link between brain and behavioral measurements would also be preserved, albeit with modulations to the strength of the correlations.

What other visual properties might correlate with cortical magnification around the visual field? It is likely that properties for which perceptual polar angle asymmetries exist, and for which V1 neurons are tuned, could also correlate with cortical magnification; for example, acuity[3,24] and spatial frequency preference[91].

We have quantified the relation between contrast sensitivity and V1 surface area, measured as a function of polar angle, across the same observers. Our data showed that: First, observers with greater contrast sensitivity tended to have a larger V1, and vice versa. Second, local contrast sensitivity can be predicted by local V1 surface area using measurements taken from the polar angle meridians. Third, a stronger horizontal-vertical asymmetry in contrast sensitivity correlated with the corresponding asymmetry in the distribution of local V1 surface area. The vertical meridian asymmetry in contrast sensitivity did not correlate with the corresponding asymmetry in the distribution of local V1 surface area, likely due to fMRI measurement constraints. Overall, these findings show that individual differences in contrast sensitivity can be linked to individual differences in V1 surface area at global and local scales and reveal that perceptual polar angle asymmetries are rooted in the cortical anatomy of V1. More

broadly, our findings show that there is a tight link between visual perception and the idiosyncratic organization of V1.

## Methods

**Observers.** 29 observers (18 females, 11 males, mean age = 29.9 years, including two authors: MMH and JW) were recruited from New York University. All observers had normal or corrected-to-normal vision and completed two experimental sessions: a 1-h psychophysics session and a 1–1.5-h fMRI scanning session. All observers provided written informed consent. The experiment was conducted in accordance with the Declaration of Helsinki and was approved by the New York University ethics committee on activities involving human observers.

**Psychophysics experiment: measuring contrast sensitivity around the visual field.** The methods used here are identical to the methods used to acquire the behavioral data in our previous study[12]. The behavioral data for 9 of the 29 observers reported here are the same as we reported in the baseline condition of that study.

**Apparatus and set up.** Observers completed the psychophysics session in a darkened, sound-attenuated room. Stimuli were presented on a 21-inch ViewSonic G220fb CRT monitor (1280 × 960 resolution, 100 Hz) and were generated using an Apple iMac (3.2 GHz, Intel Core i3) in MATLAB 2017a (Mathworks, Natick, MA) using the MGL Toolbox[92]. The CRT monitor was gamma-corrected with a ColorCal MKII colorimeter (Cambridge Research Systems, Rochester, Kent, UK) to ensure a linear relation between the frame buffer values and screen luminance. Observers viewed the monitor display binocularly with their head stabilized in a chin rest that was 57 cm from the monitor. An EyeLink 1000 eye tracker (SR Research, Ottawa, Ontario, Canada) with a sampling rate of 1000 Hz and the EyelinkToolbox[93] were used to measure fixation throughout the experiment.

**Stimuli.** The psychophysical stimulus was a Gabor patch (a sinusoidal grating embedded in a Gaussian envelope) presented on a uniform gray background. The Gabor was presented at four polar angle locations (0° on the right horizontal meridian, 90° on the upper vertical meridian, 180° on the left horizontal meridian, and 270° on the lower vertical meridian) at 4.5° eccentricity from the center of the display. The Gabor patch had a spatial frequency of 4 cycles per degree (cpd) and covered 3° of visual angle ($\sigma = 0.43°$). The Gabor patch was oriented ±15° from vertical; the orientation of the stimulus changed trial-by-trial. The contrast of the stimulus also changed trial-by-trial, based on the titration procedure.

In the center of the display was a black fixation across (0.5°). A set of four stimulus placeholders consisting of four small black squares (each 4 pixels in size) were placed just above, below, to the left, and to the right of the four locations where the Gabor patch could appear. These placeholders were included to remove spatial uncertainty about the locations the Gabor patch could appear. The placeholders remained on the screen throughout the entire experiment.

**Experimental design.** Contrast thresholds were measured at each of the four locations. To do so, observers completed a two-alternative forced choice (2AFC) orientation discrimination task in which the Gabor patch contrast was titrated using four randomly interleaved 3-down 1-up staircases via parameter estimation by sequential testing (PEST[94]). The staircases converged at 79.4% performance accuracy threshold. The staircases were interleaved across trials, and one staircase was dedicated to each of the four visual field locations. A schematic and description of a single trial is presented in Fig. 6.

In accordance with PEST rules, the Michelson contrast of the Gabor patch was titrated across trials[95]. Any trials in which the observer broke fixation (eye movements >1.5° from fixation, on average, 2.7% per block) were aborted and repeated at the end of the experimental block so that all blocks contained an equal number of trials. Observers were allowed to break fixation and blink during the response period and the ITI.

Five blocks, each consisting of 200 trials (50 trials per location), were acquired for each observer. During each block, observers were given a 20 s break after 100 trials before completing the remaining trials. Prior to completing the experiment, observers completed 1 block of 24 practice trials in which the Gabor stimuli were set to 100% contrast to ensure they were comfortable with the orientation discrimination task, eyetracking, and stimulus timing.

The 79.4% performance accuracy contrast thresholds at each of the four locations were averaged across the five independent blocks. Contrast sensitivity values were calculated as the reciprocal of these thresholds.

**fMRI experiment: retinotopic mapping and defining wedge-ROIs on the visual field meridians.** Each observer completed a 1–1.5-h scan session in which they participated in a retinotopic mapping experiment to measure population receptive field (pRF) polar angle and eccentricity estimates across visual cortex. These estimates were then used to delineate each observer's V1 map and to calculate the surface area of ±15° wedge-ROIs centered on the same angular locations as the contrast sensitivity measurements.

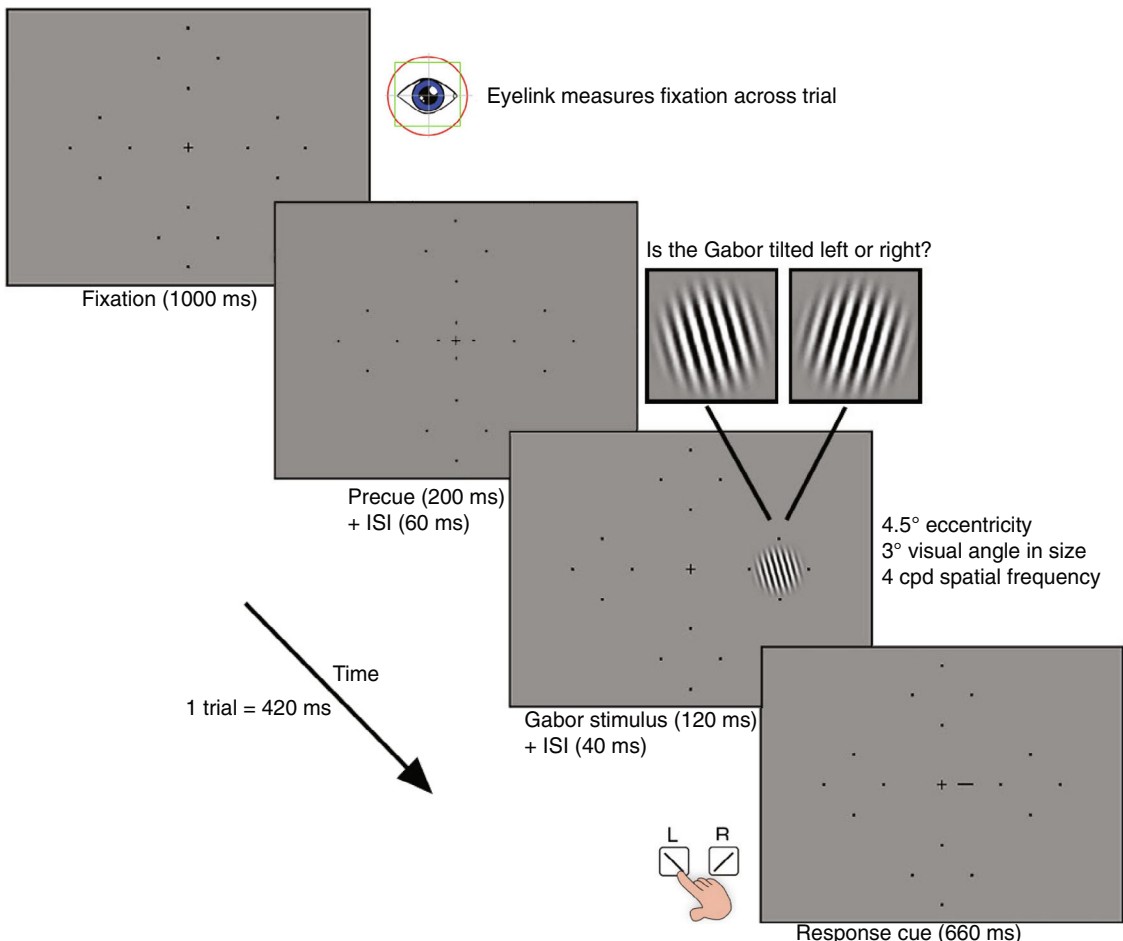

**Fig. 6 Design of the psychophysical task to measure contrast sensitivity.** Orientation discrimination task to measure contrast sensitivity at the four polar angle locations. Each trial begins with 1000 ms of fixation measured via an EyeLink eye tracker. This is followed by a 200 ms pre-stimulus cue to indicate the onset of the trial and a 60 ms inter-stimulus-interval (ISI). A Gabor patch is then presented at one of the four possible polar angle locations for 120 ms. The Gabor patch is tilted either left or right (±15°) from vertical. The offset of the Gabor patch is followed by a 40 ms ISI. A response cue (a small line indicating the location at which the Gabor patch appeared) is then presented on the screen for 660 ms. This response cue was used to eliminate uncertainty regarding the target location at very low contrasts. A brief auditory tone signaled that the observer had 5000 ms to respond, via the keyboard, as to whether the Gabor patch was tilted left or right from vertical. Auditory feedback was provided in the form of a tone to inform the observer as to whether their response was correct or incorrect. This was followed by a 1000 ms ITI before the beginning of the next trial.

The pRF stimulus, MRI and fMRI acquisition parameters, MRI and fMRI preprocessing, and the implementation of the pRF model are identical to those in our prior work[9]. The retinotopic data for 17 of the 29 observers reported here are the same as reported in that study.

**fMRI stimulus display**. Observers viewed a pRF stimulus from inside the MRI scanner bore using a ProPixx DLP LED Projector (VPixx Technologies Inc., Saint-Bruno-de-Montarville, QC, Canada). The pRF stimulus was projected onto an acrylic back-projection screen (60 cm × 36.2 cm) within the scanner bore. The projected image had a resolution of 1920 × 1080 and a refresh rate of 60 Hz. The display was calibrated using a linearized lookup table and the display luminance was 500 cd/m². Observers viewed the screen at a distance of 83.5 cm (from eyes to the screen) using an angled mirror that was mounted on the head coil.

**pRF stimulus**. Retinotopic maps were measured using pRF mapping[96]. The pRF stimulus was generated on an iMac computer using MATLAB 2017a and was projected onto the fMRI stimulus display in the scanner bore using the Psycho-physics Toolbox v3[97] and custom *vistadisp* software[98].

The pRF stimulus consisted of image patterns presented within a bar aperture that swept across the screen for the duration of each scan. The image patterns consisted of colorful objects, faces, and scenes at multiple scales that were superimposed on an achromatic pink noise $(1/f)$ background[40,99]. The image pattern was windowed within a circular aperture that had a radius of 12.4°. The image pattern was revealed through a bar aperture (3.1° wide, or 1/8th of the full stimulus extent) that swept across the screen in 24 equal steps (once per second). Each step was synchronized to the MR image acquisition (TR 1 s). There were eight

sweeps in total. Each sweep began at the edge of the circular aperture. Horizontal and vertical sweeps covered the entire diameter of the circular aperture. Diagonal sweeps only traversed half of the circular aperture; the second half of these sweeps were replaced with a blank gray screen. Each directional sweep lasted 24 s. The full stimulus run lasted 192 s. The stimulus image updated three times per second without intermediate blanks (3 Hz).

The bar aperture was superimposed on a polar fixation grid placed upon a uniform gray background, with a red or green dot at the center (3 pixels, or 0.07°). Observers were instructed to maintain fixation throughout the entire scan and completed a fixation task in which they were required to respond, using a button box, when the fixation dot changed from green to red, or vice versa.

The full stimulus sequence was completed once per functional scan. The identical aperture sequence was shown in each of the scans.

**Anatomical and functional data acquisition**. Anatomical and functional data were acquired on a 3T Siemens MAGNETOM Prisma MRI scanner (Siemens Medical Solutions, Erlangen, Germany) using a Siemens 64-channel head coil. A T1-weighted (T1w) MPRAGE anatomical image was acquired for each observer (TR, 2400 ms; TE, 2.4 ms; voxel size, 0.8 mm³ isotropic; flip angle, 8°). This anatomical image was auto-aligned to a template to ensure a similar slice prescription for all observers. Between 4 and 12 (9× had 12 scans, 2× had 11 scans, 2× had 10 scans, 5× had 8 scans, 1× had 7 scans, 8× had 6 scans, 1× had 5 scans, and 1× had 4 scans) functional echo-planar images (EPIs) were acquired for each observer using a T2*-weighted multiband EPI sequence (TR, 1000 ms; TE, 37 ms; voxel size, 2 mm³; flip angle, 68°; multiband acceleration factor, 6; phase-encoding, posterior-anterior)[100,101]. Two distortion maps were also acquired to correct susceptibility

distortions in the functional images: one spin-echo image with anterior-posterior (AP) phase encoding and one with posterior-anterior (PA) phase encoding.

**Preprocessing of structural data.** fMRIPrep v.20.0.1[102,103] was used to pre-process anatomical and functional data. For each observer, the T1w anatomical image was corrected for intensity inhomogeneity and then skull stripped. The anatomical image was automatically segmented into cerebrospinal fluid, cortical white matter, and cortical gray matter using fast[104]. Cortical surfaces were reconstructed using Freesurfer's recon-all[105] and an estimated brain mask was refined using a custom variation of the method.

**Preprocessing of functional data.** The following preprocessing was performed on each observer's functional data. First, a reference volume (and a skull stripped version) was generated using custom methodology of fMRIPrep. The two spin echo images with opposing phase-encoding directions (i.e., AP and PA distortion maps) were used to estimate a B0-nonuniformity map. The estimated distortion of the B0-nonuniformity map was then used to generate a corrected functional reference image. This corrected functional reference was co-registered to the anatomical image using six degrees of freedom.

Next, head-motion parameters with respect to the functional reference were estimated before any spatiotemporal filtering. Each functional image was slice-time corrected with all slices realigned to the middle of each TR. The slice-time corrected functional data were then resampled to the T1w anatomical space via a one-shot interpolation consisting of all the pertinent transformations (i.e., head-motion transform matrices, susceptibility distortion correction). These preprocessed time-series data were then resampled to the *fsnative* surface by averaging across the cortical ribbon.

**Implementing the pRF model to produce retinotopic maps.** The pRF model was implemented on the *fsnative* surface of each observer. For each *fsnative* vertex, the time-series data across each functional scan were averaged together to generate an average time series. These average time-series were then transformed to BOLD percent signal change (i.e., % change at each TR from the mean signal across all TRs). The pRF model was fit to the BOLD signal change.

The pRF model was implemented using *vistasoft* (https://vistalab.stanford.edu/software/, Vista Lab, Stanford University) and customized code to run the model on the cortical surface. A pRF was modeled as a circular 2D-Gaussian that was parameterized by values for $x$, $y$, and $\sigma$. The $x$ and $y$ parameters specify the center position of the 2D-Gaussian in the visual field, whereas the $\sigma$ parameter, the standard deviation of the 2D-Gaussian, specifies the size of the receptive field. The 2D-Gaussian was multiplied pointwise by the stimulus contrast aperture and was then convolved with a hemodynamic response function (HRF) to predict the BOLD percent signal change. We parameterized the HRF by five values, describing a difference of two gamma functions[64,96,106,107].

The pRF model was implemented using a coarse-to-fine approach to find the optimal $x$, $y$, and $\sigma$ for each vertex by minimizing the residual sum of squares between the predicted time-series and BOLD signal[96]. The $x$ and $y$ values were then used to calculate vertex-wise eccentricity and polar angle coordinates, reflecting the pRF center position in the visual field. All analyses were completed using data with $R^2 = >10\%$.

**Defining V1.** V1 was defined as a region-of-interest (ROI) by hand using Neuropythy v0.11.9 (https://github.com/noahbenson/neuropythy[108]). We defined V1 from 0° to 8° eccentricity with the V1/V2 dorsal border falling through the center of the lower vertical meridian, and the V1/V2 ventral border falling through the center of the upper vertical meridian.

**Cortical magnification as a function of polar angle analysis.** To calculate local measurements of V1 surface area along the polar angle meridians, we defined ±15° wedge-ROIs that were centered along each of the four polar angle meridians in the visual field. We measured the amount of V1 surface area within these wedge-ROIs[9]. As the wedge-ROIs at each meridian are always defined to encapsulate ±15° of visual space, any differences in the amount of localized V1 surface area calculated using these wedge-ROIs can be interpreted as a measurement of cortical magnification.

The specific implementation of the cortical magnification analysis is described below and is taken from our prior work[9].

In brief, we defined ±15° wedge-ROIs that were centered on the cardinal meridians: the left and right horizontal meridian, and the upper and lower vertical meridians in the visual field. Each wedge-ROI extended from 1° to 8° eccentricity. Unlike the 0–8° limit we used for mapping V1 size, we excluded the central 1° from the wedge-ROIs because the polar angle representations can be relatively noisy around the foveal representation and this can impact the border definition of the wedge-ROI[4,109]. We implemented a method of defining these wedge-ROIs to avoid the speckles and discontinuities that would inevitably arise had we simply defined the ROIs by estimates of pRF centers[9,108]. To define a 15° wedge-ROI, we first calculate several sub-wedge-ROIs that are constrained to a narrow eccentricity band. Each eccentricity band extends 15° from a meridian. The location of the 15° border of the wedge-ROI on the cortex is calculated using the average distance (in

mm) of a pool of vertices whose pRF polar angle coordinates lie near the edge of the 15° boundary in visual space. The eccentricity-defined sub-wedge-ROIs are concatenated to form one full wedge-ROI. We then use the wedge-ROI as a mask and calculate the surface area of the vertices falling within the wedge-ROI.

**Defining wedge-ROIs.** For each observer, we first computed the shortest distance on the *fsnative* surface between each pair of vertices and a cardinal meridian. For each observer, we defined the horizontal, upper, and lower vertical meridians of V1 using manually defined line-ROIs. These line-ROIs were defined on the flattened *fsnative* surface using Neuropythy[108]. The meridian definitions were informed by anatomy (i.e., the curvature map), pRF data (polar angle maps), and the V1 border. The line-ROIs were used to generate three *cortical distance maps* (one for the upper vertical meridian, one for the lower vertical meridian, and one for the horizontal meridian), again using Neuropythy. This was completed for the left and right hemispheres of V1. The cortical distance maps specify the distance of each vertex from the respective cardinal meridian (in mm), with the distance of the meridian itself set to 0 mm.

Next, we divided the V1 map into 10 eccentricity bands that were log spaced between 1° and 8° of eccentricity in the visual field to be approximately equally spaced on the cortex. These eccentricity bands were used to generate sub-wedge-ROIs that are later combined to form the full wedge-ROI.

Within each eccentricity band, we used the cortical distance maps described above to compute the distance (in mm) of a 15° iso-angle line from the center of the wedge (i.e., a visual field meridian, whose distance is by default 0 mm). This iso-angle line represents the outer boundary of the wedge-ROI in the visual field. The distance of the 15° iso-angle line from a meridian was identified by calculating the average distance of the vertices in a region of cortex that fell ±8° "around" the 15° iso-angle line, using the polar angle values derived from the pRF data. The average distance of these vertices outputs a value that represents the cortical distance (in mm) of the 15° iso-angle line from a meridian. This process was repeated for each eccentricity band.

For each eccentricity band, we identified the vertices that had cortical distance value between 0 mm (i.e., falling along a meridian) and the distance of the 15° iso-angle line ($x$ mm, based on the calculation above). This process was repeated to create sub-wedge-ROIs at each eccentricity band. The sub-wedge-ROIs are combined to form a full wedge-ROI mask that extends out to 15° in width from a meridian and ranges between 1° and 8° of eccentricity. This process is repeated for each meridian, and each hemisphere (to form the opposing portion of the full ±15° wedge-ROI).

Finally, we overlaid the wedge-ROI mask on cortical surface area maps. The cortical surface area maps are generated for each observer using Neuropythy and specify the cortical surface area (in mm²) of each vertex on each obervers *fsnative* surface. The cortical surface area of the wedge-ROI is calculated by summing the surface area of the vertices within the wedge-ROI mask. This outputs the total surface area of a 15° wedge-ROI in one hemisphere. For each wedge-ROI, the surface area from the left and right hemisphere of V1 are summed together to calculate the surface area for the horizontal meridian (across the left and right visual field), the upper vertical meridian, and the lower vertical meridian, effectively forming ±15° wedge-ROIs that extend 15° either side of each meridian. The surface areas of the upper and lower vertical meridian are summed to calculate the surface area of the full vertical meridian.

**Midgray, pial, and white matter surfaces.** We assessed the cortical surface area using surface area maps generated at three different depths—midgray surface, pial surface, and white matter surface. These maps are generated using *Freesurfer*. The main analyses were conducted using the midgray cortical surface. The supplemental analyses used pial and white matter cortical surfaces. This is because the surface area changes as a function of cortical depth. The surface area of gyri at the pial surface and the sulci at the white matter surface tend to be large, whereas the reverse (sulci at the pial surface and gyri at the white matter surface) are smaller.

**Summary to relate psychophysical and fMRI analyses.** Figure 7 summarizes how the contrast sensitivity measurements relate to the wedge-ROIs measurements of local V1 surface area. Contrast sensitivity was measured at four polar angle locations: the left and right horizontal, upper vertical, and lower vertical meridians (Fig. 7a). Contrast sensitivity measurements were made using Gabor patches that were 3° of visual angle in size and were placed at 4.5° eccentricity from fixation. Next, we calculated the pooled surface area of V1 vertices falling within ±15° wedge-ROIs that were centered along the horizontal (green mask), upper vertical (blue mask) and lower vertical (red mask) meridians (Fig. 7b, left hemisphere V1/right visual hemifield). These measurements are calculated for the left and right hemisphere (thus the wedge-ROIs were centered on each meridian and extended 15° either side. As the upper and lower vertical meridian is split across hemispheres, the ±15° wedge-ROIs for these meridians were formed by combining the V1 surface area of the 15° wedges that encapsulated data from the left and right portions of V1.

Therefore, we calculated measures of cortical magnification along the polar angle meridians that matched the angular location of contrast sensitivity measurements. Note that we do not intend for our cortical magnification

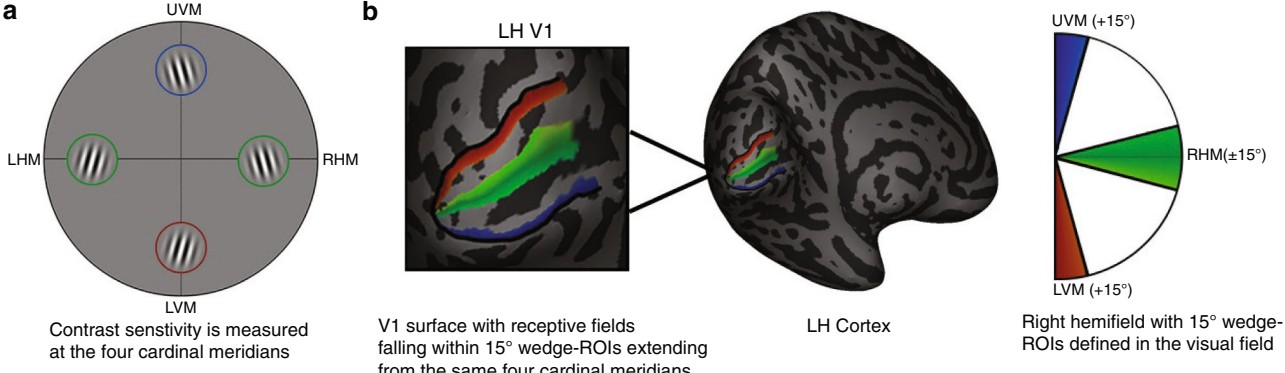

**Fig. 7 Schematic of experiment. a** Contrast sensitivity is measured at the four polar angle meridians using an orientation discrimination task. **b** 15° wedge-ROIs are defined in the visual field and extend from the four polar angle meridians that match the angular location of the contrast sensitivity measurements. The V1 surface area dedicated to processing the regions of visual space within the wedge-ROI mask is calculated. The surface area of each wedge-ROI is summed across the left and right hemispheres to form a ±15° wedge centered on each meridian.

measurements to be an exact measurement of the amount of V1 surface area dedicated to processing the Gabor stimulus itself (i.e., a 3° × 3° patch at 4.5° of eccentricity in V1 at each polar angle meridian). Measurements of cortical magnification are noisy, and the more data included in the calculation of the surface area of the wedge-ROI, the more accurate the output value. We chose to measure the surface area of wedge-ROIs ±15° of width in angle and extending out to 8° of eccentricity when the Gabor patches in the visual field extend ±1.5° either side of each polar angle meridian and were centered at 4.5° of eccentricity. This is because there is a tradeoff in the size of the wedge-ROI and the accuracy of cortical magnification measurements (especially along the vertical meridian where data is comparatively sparse when compared to the horizontal meridian)[9].

**Reporting summary**. Further information on research design is available in the Nature Research Reporting Summary linked to this article.

## Data availability

Source data are provided with this paper. The data generated for this study have been deposited in the OSF repository https://osf.io/de7zg. Source data are provided in this paper.

## Code availability

Scripts used for data collection and analysis code to generate manuscript figures are available in the OSF repository at https://osf.io/de7zg and on GitHub at https://github.com/WinawerLab/vistadisp.

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

## Acknowledgements

We thank Michael Jigo, Eline Kupers, and Ekin Tünçok for their feedback on the manuscript and Noah Benson for software support. This research was funded by the US NIH National Eye Institute R01-EY027401 and P30-EY-013079 to M.C. and J.W.

## Author contributions

M.M.H., J.W., and M.C. designed the experiments, M.M.H. performed the experiments, M.M.H. and J.W. analyzed the data, M.M.H, J.W., and M.C. wrote and edited the paper, and J.W. and M.C. acquired funding.

## Competing interests

The authors declare no competing interests.
