## [Peer Review File · Nature Communications]

Linking individual differences in human primary visual cortex
to contrast sensitivity around the visual fieldREVIEWER COMMENTS

Reviewer #1 (Remarks to the Author):

REVIEW FOR HIMMELBERG ET AL. 'LINKING INDIVIDUAL DIFFERENCES IN HUMAN V1 TO PERCEPTION AROUND THE VISUAL FIELD' NCOMMS-21-44660

SUMMARY

This paper tests a longstanding hypothesis regarding the neural basis of visual field anisotropies in human contrast sensitivity. Namely that local contrast sensitivity depends on the pooling of local neural activity and therefore scales with the number of primary visual cortex neurons representing a given visual field position. This pooling hypothesis was first derived from the observation that – on average - contrast sensitivity (and acuity) falls off with eccentricity in a manner paralleling the general cortical magnification factor (Virsu & Romano, 1979). Later studies (including some by the authors) have shown that this type of match between cortical magnification and contrast sensitivity (as well as other perceptual effects) extends to well-known polar angle anisotropies. Both, contrast sensitivity and V1 cortical magnification are higher along the horizontal compared to the vertical meridians (the horizontal-vertical anisotropy, HVA) and higher for the lower compared to the upper vertical meridian (the vertical meridian anisotropy, VMA). These observations are in line with the pooling hypothesis, but are at least in part post-hoc explanations of previously observed anisotropies.

Here, the authors put the pooling hypothesis to a novel and strong test using a straightforward prediction. Besides the well-known common anisotropies in cortical magnification, the overall size of V1 as well as local cortical magnification varies substantially from individual to individual. Under the pooling hypothesis, this interindividual variance in functional neuroanatomy should predict individual differences in local contrast sensitivity. Using functional MRI, population receptive field mapping and psychophysics the authors found this indeed to be the case, at least for individual variation in overall V1 size (up to 8 deg ecc.) and overall contrast sensitivity as well as for the neural and perceptual HVA.

EVALUATION

Reading this paper was a joy. It is exceptionally well written and presents convincing evidence regarding a straightforward, novel prediction, putting an important and longstanding theory to the test. The methods are solid, the discussion is nuanced and the conclusions are warranted by the data. I strongly recommend publication, but have a number of minor suggestions, which I will detail below.

SUGGESTIONS

-> TITLE. I suggest changing the title to something more specific, such as 'Linking individual differences in human V1 to contrast sensitivity around the visual field'. As far as I see, there are at least three previous reports of local V1 area correlating with individual perceptual anisotropies (two of which are cited in the manuscript; also see below: Duncan & Boynton, 2003, Neuron; Song et al., 2015 Neuron; Moutsiana et al., 2016 Nature Comms).

-> BOOTSTRAPPING. 1k iterations seem a bit low, I would suggest using 10k to be on the safe side. Also, it may be more intuitive to report 'p-values' derived from the proportion of random effects greater/smaller than the one observed empirically. More importantly, I think the explanations of the analyses in lines 228ff and lines 259ff need to become a bit more detailed and clearer, including an explanation of which statistical hypothesis is tested in which instance.

I *think* in both cases the shuffling was only between observers, but not within (i.e. the ties between *elements* of a given neural or behavior quadruple were not broken, but the pairing between these

neural and behavioral quadruples was). Is this correct?

If so, this would mean the analysis in lines 228ff specifically tests the significance of the relationship between individual differences in cortical magnification and contrast sensitivity *over and above the general (shared) HVA and VMA effects* and *including both, individual variance in overall V1 size and local anisotropy*, correct? If so, I think this should be spelled out more clearly.

Relatedly, the analysis in lines 259ff tests the significance of individual differences in anisotropy specifically *when excluding effects of overall V1 size*, correct? The high correlations in the bootstrapped 'null distribution' reflect the general (i.e. shared) component of HVA and VMA, which was not removed, correct? If so, the reason Figure 5a is interpreted to indicate a 'robust effect of polar angle' (line 264f) despite the observed correlation falling just within the bootstrapped 95% distribution is that the sentence refers to the general effect, not individual variation, correct?

I hope and suspect I got the right idea on all of these from the text, but feel a bit more unpacking could greatly help the reader. This is probably just a matter of adding a couple of explanatory sentences.

-> Interpretation of ns VMA CORRELATION. I think the sentence in line 301ff is a bit misleading. The *individual* VMA in contrast sensitivity and cortical magnification were not correlated – but that doesn't mean the *general* asymmetries weren't. This could simply be fixed by more precise phrasing, such as 'Thus, the *individual* asymmetry... not associated with the *individual* amount of...'.
.

Relatedly, the following sentence in line 450 is a bit cryptic and could do with some unpacking: 'fMRI measurements are more difficult near areal boundaries, thereby impacting the precision of vertical meridian measurements at the individual level.' What exactly do you mean? I guess an important factor is the inherent ambiguity in delineating the VM borders exactly and deciding which of the vertices fall on the V2 side and which are part of V1?

-> HISTOLOGY. The paragraph in lines 378ff at times seems circular. 'The fact that performance on at least some tasks correlates with V1 surface area^{42–45} implies that a smaller V1 likely has fewer overall neurons' - isn't that the hypothesis you set out to test? It would seem more careful to replace 'implies' with 'suggest' or similar. Also, smaller pRFs in larger V1s only suggest finer sampling of visual space under the assumption of constant neural density – if neuronal spacing varies, the scatter of pRFs per unit surface area could vary without any sampling differences (imagine the same grid of sampling points stretched). I think these arguments are good ones to make, but adding a 'may' or similar would be more appropriate given we currently can't escape the open question the par begins with.

-> PRIOR STUDIES. Lines 406ff claim 'The only prior study relating individual differences in cortical magnification to perceptual outcomes reported a positive correlation between V1 cortical magnification and visual acuity measured as a function of eccentricity' citing Duncan & Boynton, 2003. I think there are at least two more examples: intraindividual variation in local V1 area across polar angles or quadrants has been linked to local position discrimination ability by Song et al, 2015 (their figure 4) and to subjective size perception biases by Moutsiana et al. 2016 Nat Comms (their figure 5). Also, Song et al., 2013 Nat Comms found no correlation between overall V1 size and contrast discrimination threshold. Please briefly discuss this apparent discrepancy to your findings.

-> METHODS.

Did you apply any threshold when considering pRF data, such as a minimum percentage of variance explained by the model?

Lines 681ff wedge ROI definition – Please give more detail on how this worked in practice. Figure 8 looks like the ROI masks were continuous. How was this achieved? How did you deal with the (I

guess inevitable) holes/discontinuities caused by noise in the maps? Did you delineate the wedge mask by hand based on a binary projection of vertices with a matching polar angle preference?

Please give descriptive stats on the number of aborted trials due to lacking fixation compliance in the behavioral experiment.

Please report the consistency of individual differences in contrast sensitivity anisotropies across blocks of the behavioral experiment (such as split half correlations).

Just to double-check: did you ask ppts for red/green deficiencies / did you check their performance for the fixation task in the scanner?

-> IMPLICATIONS AND FUTURE RESEARCH. Were other perceptual tests conducted? Why (not)? Which perceptual properties should future studies consider? It may be a bit speculative, but would be interesting to briefly discuss why apparently there is little evolutionary pressure towards a homogenous 'optimal' V1 size (not even in relation to the overall size of cortex). Why do we invest vastly different resources into early visual processing, even though this has clear functional consequences, such as the one you have shown?

Reviewer #2 (Remarks to the Author):

Himmelberg et al show that a basic perceptual measure, contrast sensitivity, is correlated with local and global V1 surface area. The manuscript is convincing, well-written and the data presented clearly. I have only a few remarks.

- It feels like the paper could be shorter. For instance, figure 4 and 5 in my mind belong together, as it is -in my mind- immediately necessary to tease apart the effects in figure 4 as is done in figure 5. But I would agree that this could be a matter of taste.

- The wedge surface ROI the authors use here seems to be adopted from their earlier publication on reproducibility. This is reasonable in a historical sense, but I feel as if the ROIs could have been tailored much more precisely to the exact location and extent of the stimuli used in the psychophysical experiment. Even though the polar angle extent of the wedges isn't that far off, 2 to 8 degrees eccentricity feels a bit coarse in its correspondence with the stimuli. I understand that one might want to fit a CSS and not a linear pRF model to predict the surface pattern of activation and that the extent would also depend on simulated contrast, but wouldn't the authors agree that better correspondence here should only strengthen their results? I don't want to give the authors an arbitrary hoop to jump through, and agree with their reasoning in the methods in principle. I should stress that I already approve of the work but with this addition would cause me to admire it more, and that I believe that including this analysis would mostly serve to increase uptake in the future.

- The p-value for the correlation shown in figure S3 is different from the one stated in the text. Which is the true value?

- On the same point: Fig S3 shows the correlation after correcting for cortical surface area, but could the authors also show the correlation of contrast sensitivity with total cortical surface area? Is this correlation also significant? Presenting this result would speak more clearly to the specificity of the findings.

- 'Biological sex' (page 7) seems a bit of a weird and ambiguous concept to me. I understand where the use of this convoluted concept comes from (I guess non-biological sex is self-identified gender) and don't have a preferred alternative, but biological sex could be determined by karyotype, or by phenotype (which can be dissociated). Wouldn't just 'sex' be enough? Apologies for the awkward segue.

Reviewer #3 (Remarks to the Author):

Summary

The authors have compared cortical magnification (as measured with fMRI) with visual performance (contrast sensitivity) at the four polar meridians of the visual field. In line with previous work, they show that greater cortical magnification is associated with greater visual performance. Importantly, they show that this relationship is present at the level of the individual, as well as the group. This neatly provides novel evidence connecting perception with anatomy at the level of individual observers.

This paper is well written and the findings will be of interest to a broad range of researchers of visual function. However, some analyses are not clear and should be better explained, see comments below. The discussion is rather limited and should be improved.

Major Points

[L26]: In the abstract the authors mention the correlation they found between cortical and behavioural HVA. They do not mention that correlations between cortical and behavioural VMA were also explored, but that no correlation was found. This is made clear in the discussion, and potential explanations are offered. However, it is perhaps a little misleading to include the HVA findings in the abstract, without also reporting on the VMA. Especially when the abstract concludes: "These data reveal a tight link between cortical anatomy and visual perception at the level of individual observer and stimulus location.", without a caveat.

[L264]: The authors state that "After factoring out between-observer variability, there was still a positive relation between local contrast sensitivity and local V1 surface area ($r_p = .78$, $x_{0.95} = .79$)". Here the Spearman's rho value is less than the 95th percentile of the bootstrapped distribution. The authors do not highlight this or justify why this result would nevertheless "indicate a robust effect of polar angle".

[L429]: The discussion is rather limited. For example, the authors discuss the possible origin of various asymmetries. While the role of midget ganglion cells is mentioned, there's no clear conclusion on the extent to which this determines the various reported asymmetries and individual differences in contrast sensitivity. I would prefer to see a more in-depth discussion of this.

Another issue to address: In the present study, contrast sensitivity is measured only for one stimulus size and near vertically oriented gratings at one spatial frequency. Would the present results generalize to other orientations, spatial frequencies, and stimulus sizes? Is it possible that reported patterns would change for different types of stimuli, or stimuli in context rather than on a uniform gray background.

This touches upon the stimulus used to measure the pRFs and thus to estimate cortical magnification. This stimulus contained a variety of patterns "The pRF stimulus consisted of image patterns presented within a bar aperture that swept across the screen. The image patterns consisted of colorful objects, faces, and scenes at multiple scales that were superimposed on an achromatic pink noise (1/f) background."

Please discuss whether this difference between psychophysical stimulus and pRF stimulus may affect results in any way. The surface area estimates relied on pRF estimates, which can be strongly affected by stimulus properties (e.g. DOI: 10.1146/annurev-vision-091517-033948 / DOI: 10.1016/j.neuroimage.2017.06.073)

Consider whether it would not have been better to assess cortical surface using the same stimulus as used in the psychophysics (or alternatively, somehow assess contrast sensitivity for a range of patterns or stimuli and on an achromatic noise background.

Minor Points

[L44]: "Visual performance also changes as a function of polar angle" - the term visual performance is a little vague, the authors should consider making it clear what aspects of vision they are referring to or give examples (i.e., contrast sensitivity).

[L99]: "We limit the range to 8° because the functional data are less reliable near the edge of the retinotopic mapping stimulus (12.4°)" - if this (8 deg) cutoff has been determined by previous studies, please provide a citation. If it is arbitrary or the parameters the authors have found to be useful, please state this.

[L113]: Figure 1B: It states that "The y-axes are matched so that the scaling in (B) is 4x the size of that in panel (A)." This should be 100x ($0.2 \text{ m}^2 = 200.000 \text{ mm}^2$).

[L114]: The (n=29) is positioned a bit in a confusing way, as it reads as it refers to hemispheres, whereas it concerns the # of observers.

[L113]: Figure 1D, left plot of polar angle. The two inflated cortices are placed on top of each other, and I find it difficult to distinguish where one surface starts and the other begins. It would be easier to compare the two if they did not overlap, perhaps by stacking them vertically. The same problem applies to the figure on the right of eccentricity.

[L139]: Here the authors explain how the asymmetry index is calculated, but only using text. Mathematical equations would be easier to interpret and look-up. It would also be useful to explain what the index means: i.e., how would we interpret a VMA index of 0 compared to 100? Moreover, it seems very very little is done with these indices. For example, the statistics reported next are done using the original contrast sensitivity values.

[L203]: In figure 3, cs is plotted on the x-axis, whereas surface is plotted on the y-axis. The reverse would make more sense, and would also more easily show the described non-linear relationship between surface area and cs. The inset figure is almost too tiny to be useful. Why not add a panel next to the figure to show it, if it is deemed important.

[L250]: Again, it is not clear from this paragraph (or the supplementary figures it refers to) what is being bootstrapped, and how the values in Figures S4A, S4B and S4C are different from each other and how they are calculated. I assume that for:

Figure S4A: raw CS and wedge-ROI-SA values are shuffled

Figure S4B: CS & wedge-ROI-SA values, whose between-observer variance has been removed are shuffled

Figure S4C: CS & wedge-ROI-SA values, whose within-observer variance has been removed are shuffled

This could all be made clearer.

[L378]: The authors highlight an interesting question raised by their findings: how variation in size of V1 relates to neural circuitry. They cite (47) stating "the size of V1 is inversely correlated with the size of its pRFs, suggesting that a larger V1 enables finer sampling of visual space". Given the authors calculated pRF as part of the main analyses and raised this interesting point in the discussion, it is perhaps a shame that they did not also compare wedge-ROI-surface area with pRF size. This would help answer the question they raise in this paragraph. A related general question is whether the size of their Gabors is optimal for each putative "V1 sampling size".

[L411]: Here, the authors list evidence for many asymmetries in perception. However, there's little to no attempt at discussing or integrating this information. E.g. Song et al. (2013), already cited by the authors, suggest certain trade-offs affecting surface area (spatial detail vs. contextual influences). Is there a tradeoff in which contrast sensitivity fits as well? I would like to read the author's educated guesses on this.

[L681]: How exactly are wedge-ROIs calculated from retinotopic data? By selecting vertices whose pRF centers fall inside the wedges in the visual field, or only those vertices whose pRF is entirely inside the wedges in the visual field?

[L894]: the reference listed is to a biorxiv manuscript, but there's already an accepted publication associated with it.

[L902]: Since the authors use the Psychophysics toolbox and the Eyelink, it is likely that they also use the code of the EyelinkToolbox. If so, acknowledge this by citing <https://doi.org/10.3758/BF03195489>

Linking individual differences in human primary visual cortex to contrast sensitivity around the visual field

Response to Reviewers

We would like to thank the reviewers for their constructive feedback on our manuscript. We have taken all their comments into consideration and our responses are below in blue text. Changes to the revised manuscript have been tracked in blue text and are noted below in italics with line numbers for reference. Addressing the reviews has strengthened our manuscript.

Best,

Marc Himmelberg, Postdoctoral Research Associate

Jonathan Winawer, Associate Professor of Psychology and Neural Science

Marisa Carrasco, Julius Silver Professor of Psychology and Neural Science

Reviewer 1.

1. Title. I suggest changing the title to something more specific, such as 'Linking individual differences in human V1 to contrast sensitivity around the visual field'. As far as I see, there are at least three previous reports of local V1 area correlating with individual perceptual anisotropies (two of which are cited in the manuscript; also see below: Duncan & Boynton, 2003, *Neuron*; Song et al., 2015 *Neuron*; Moutsiana et al., 2016 *Nature Comms*).

– Agreed. We have now changed the title to '*Linking individual differences in human primary visual cortex to contrast sensitivity around the visual field*'.

2. BOOTSTRAPPING. 1k iterations seem a bit low, I would suggest using 10k to be on the safe side. Also, it may be more intuitive to report 'p-values' derived from the proportion of random effects greater/smaller than the one observed empirically. More importantly, I think the explanations of the analyses in lines 228ff and lines 259ff need to become a bit more detailed and clearer, including an explanation of which statistical hypothesis is tested in which instance.

I *think* in both cases the shuffling was only between observers, but not within (i.e. the ties between *elements* of a given neural or behavior quadruple were not broken, but the pairing between these neural and behavioral quadruples was). Is this correct?

If so, this would mean the analysis in lines 228ff specifically tests the significance of the relationship between individual differences in cortical magnification and contrast sensitivity *over and above the general (shared) HVA and VMA effects* and *including both, individual variance in overall V1 size and local anisotropy*, correct? If so, I think this should be spelled out more clearly.

Relatedly, the analysis in lines 259ff tests the significance of individual differences in anisotropy specifically *when excluding effects of overall V1 size*, correct? The high correlations in the bootstrapped 'null distribution' reflect the general (i.e. shared) component of HVA and VMA, which was not removed, correct?

If so, the reason Figure 5a is interpreted to indicate a ‘robust effect of polar angle’ (line 264f) despite the observed correlation falling just within the bootstrapped 95% distribution is that the sentence refers to the general effect, not individual variation, correct? I hope and suspect I got the right idea on all of these from the text, but feel a bit more unpacking could greatly help the reader. This is probably just a matter of adding a couple of explanatory sentences.

– We have redone the bootstrapping analysis and combined Figures 4 and Figures 5 into a single **Figure 4**. We have simplified our analyses and unpacked the details of these analyses and interpretation of the results in our revised text (**Lines 227-244 and Lines 263-271**).

3. Interpretation of ns VMA CORRELATION. I think the sentence in line 301ff is a bit misleading. The *individual* VMA in contrast sensitivity and cortical magnification were not correlated – but that doesn’t mean the *general* asymmetries weren’t. This could simply be fixed by more precise phrasing, such as ‘Thus, the *individual* asymmetry... not associated with the *individual* amount of...’.

– We have corrected the sentence as suggested.

Line 318-319: *Thus, the individual asymmetry for contrast sensitivity between the lower and upper vertical meridian was not associated with the individual amount of V1 surface area dedicated to the lower and upper vertical meridian.*

4. Relatedly, the following sentence in line 450 is a bit cryptic and could do with some unpacking: ‘fMRI measurements are more difficult near areal boundaries, thereby impacting the precision of vertical meridian measurements at the individual level.’ What exactly do you mean? I guess an important factor is the inherent ambiguity in delineating the VM borders exactly and deciding which of the vertices fall on the V2 side and which are part of V1?

– We have now clarified this in the manuscript.

Lines 479-486: *Another possibility is measurement constraints; the vertical meridian lies at the extreme range of the polar angle distribution within a hemisphere. Therefore, the blurring of neural responses due to the fMRI measure will skew pRF centers away from the vertical meridian representation. Many studies have noted a lack of representation of the visual field close to the vertical meridian in V1 and other visual field maps, presumably for this reason ^{40,86–88}. This limitation introduces some noise into estimates of the size of the cortical representation of the vertical meridian, compounding with the reduced SNR due to an overall smaller representation of the vertical than horizontal meridian.*

5. HISTOLOGY. The paragraph in lines 378ff at times seems circular. ‘The fact that performance on at least some tasks correlates with V1 surface area implies that a smaller V1 likely has fewer overall neurons’ - isn’t that the hypothesis you set out to test? It would seem more careful to replace ‘implies’ with ‘suggest’ or similar. Also, smaller pRFs in larger V1s only suggest finer sampling of visual space under the assumption of constant neural density – if neuronal spacing varies, the scatter of pRFs per unit surface area could vary without any sampling differences (imagine the same grid of sampling points stretched). I think these arguments are good ones to make, but adding a ‘may’ or similar would be more appropriate given we currently can’t escape the open question the par begins with.

– We have now amended the manuscript to soften our language.

Lines 402-404: *The fact that performance on some tasks correlates with V1 surface area^{52,55-57} suggests that a smaller V1 likely has fewer overall neurons, but the histological measures to directly assess this do not yet exist*

6. PRIOR STUDIES. Lines 406ff claim ‘The only prior study relating individual differences in cortical magnification to perceptual outcomes reported a positive correlation between V1 cortical magnification and visual acuity measured as a function of eccentricity’ citing Duncan & Boynton, 2003. I think there are at least two more examples: intraindividual variation in local V1 area across polar angles or quadrants has been linked to local position discrimination ability by Song et al, 2015 (their figure 4) and to subjective size perception biases by Moutsiana et al. 2016 Nat Comms (their figure 5). Also, Song et al., 2013 Nat Comms found no correlation between overall V1 size and contrast discrimination threshold. Please briefly discuss this apparent discrepancy to your findings.

– We have now amended this section.

Lines 427-431: *Indeed, three studies have related individual differences in localized measurements of cortical magnification to perceptual outcomes. Local V1 cortical magnification positively correlates with visual acuity measured as a function of eccentricity³, position discrimination ability at different angular locations⁵⁵, and subjective object size for different visual field quadrants⁶⁵.*

– We now briefly discuss differences between Song et al (2013) findings and ours.

Lines 386-388: *V1 surface area has been shown to correlate with a few measurements of visual performance, such as perceptual acuity thresholds^{55,56}, measurements of subjective object size⁵², and orientation discrimination thresholds⁵⁷, but not with contrast discrimination thresholds⁵⁷. Such thresholds are different from those measured here; they depend upon the range of the contrast response function being measured^{58,59}, but not upon stimulus orientation.*

7. Did you apply any threshold when considering pRF data, such as a minimum percentage of variance explained by the model?

– Yes, clarified.

Line 718: *All analyses were completed using data with $R^2 \Rightarrow 10\%$.*

8. Lines 681ff wedge ROI definition – Please give more detail on how this worked in practice. Figure 8 looks like the ROI masks were continuous. How was this achieved? How did you deal with the (I guess inevitable) holes/discontinuities caused by noise in the maps? Did you delineate the wedge mask by hand based on a binary projection of vertices with a matching polar angle preference?

– We have expanded our Methods section on how the wedge-ROIs worked in the manuscript (see new Methods section *Cortical magnification as a function of polar angle analysis* on **Lines 728-797**).

9. Please give descriptive stats on the number of aborted trials due to lacking fixation compliance in the behavioral experiment.

– Please note that any aborted trials were repeated, so all blocks have the same number of observations.

Lines 581-583: *Any trials in which the observer broke fixation (eye movements > 1.5° from fixation, on average, 2.7% per block) were aborted and repeated at the end of the experimental block so that all blocks contained equal trials*

10. Please report the consistency of individual differences in contrast sensitivity anisotropies across blocks of the behavioral experiment (such as split half correlations).

– We have done so.

Lines 151-154: *Both the HVA and VMA were well-correlated across subsampled blocks of the behavioral data from each observer, indicating that the contrast sensitivity measurements were reliable within each observer and supporting the use of contrast sensitivity measures as reflecting genuine individual differences (Supplementary Fig. 1).*

11. Just to double-check: did you ask ppts for red/green deficiencies / did you check their performance for the fixation task in the scanner?

– We didn't specifically check for any red/green deficiencies, but all participants informed us they had normal or corrected to normal vision and they could do the color discrimination task. Performance on this task is recorded but not reported. We use a camera in the scanner to watch the participant live and ensure that they are awake.

12. IMPLICATIONS AND FUTURE RESEARCH. Were other perceptual tests conducted? Why (not)? Which perceptual properties should future studies consider? It may be a bit speculative, but would be interesting to briefly discuss why apparently there is little evolutionary pressure towards a homogenous 'optimal' V1 size (not even in relation to the overall size of cortex). Why do we invest vastly different resources into early visual processing, even though this has clear functional consequences, such as the one you have shown?

– We now discuss this.

Lines 504-507: *What other visual properties might correlate with cortical magnification around the visual field? It is likely that properties for which perceptual polar angle asymmetries exist, and for which V1 neurons are tuned, could also correlate with cortical magnification; for example, acuity^{3,24} and spatial frequency preference⁸⁹.*

– We have added a brief comment noting variability in earlier stages of the visual system might contribute to V1 size.

Lines 369-372: *Why is there so much variability in the size of V1? One hypothesis is that variation in the size of V1 depends on the amount of detail encoded in earlier stages of the visual system: cone density varies by about 3-fold across observers⁵³ and the size of the LGN and optic tract also vary substantially^{1,54}*

Reviewer 2.

1. It feels like the paper could be shorter. For instance, figure 4 and 5 in my mind belong together, as it is -in my mind- immediately necessary to tease apart the effects in figure 4 as is done in figure 5. But I would agree that this could be a matter of taste.

– Agreed. We now combine Figures 4 and Figure 5 into a single **Figure 4** with an updated caption. In addition, we have tightened the manuscript throughout.

2. The wedge surface ROI the authors use here seems to be adopted from their earlier publication on reproducibility. This is reasonable in a historical sense, but I feel as if the ROIs could have been tailored much more precisely to the exact location and extent of the stimuli used in the psychophysical experiment. Even though the polar angle extent of the wedges isn't that far off, 2 to 8 degrees eccentricity feels a bit coarse in its correspondence with the stimuli. I understand that one might want to fit a CSS and not a linear pRF model to predict the surface pattern of activation and that the extent would also depend on simulated contrast, but wouldn't the authors agree that better correspondence here should only strengthen their results? I don't want to give the authors an arbitrary hoop to jump through, and agree with their reasoning in the methods in principle. I should stress that I already approve of the work but with this addition would cause me to admire it more, and that I believe that including this analysis would mostly serve to increase uptake in the future.

– We note that our analysis has greater spatial specificity than previous work for which cortical surface measurements are related to behavior. The more data included in the wedge-ROI, the better the cortical magnification estimate. This is especially important for individual differences. Thus, there is a necessary compromise between matching the wedge-ROI size to our relatively small psychophysical stimulus (which will *increase the noise* in the cortical magnification measurement) and making the wedge-ROI larger but less well matched to the psychophysical stimulus (which will *decrease the noise* of the cortical magnification measurement). We are continuously developing the software and methods with the goal of improving the precision of cortical magnification with less compromise.

3. On the same point: Fig S3 shows the correlation after correcting for cortical surface area, but could the authors also show the correlation of contrast sensitivity with total cortical surface area? Is this correlation also significant? Presenting this result would speak more clearly to the specificity of the findings.

– We have now run this new correlation.

Line 214-216: *Further, average contrast sensitivity did not correlate with overall cortical surface area ($p > .1$), indicating that these correlations are specific to V1.*

4. 'Biological sex' (page 7) seems a bit of a weird and ambiguous concept to me. I understand where the use of this convoluted concept comes from (I guess non-biological sex is self-identified gender) and don't have a preferred alternative, but biological sex could be determined by karyotype, or by phenotype (which can be dissociated). Wouldn't just 'sex' be enough? Apologies for the awkward segue.

– We have amended this to say, 'not a result of sex differences' (**Line 116**).

Reviewer 3.

1. [L26]: In the abstract the authors mention the correlation they found between cortical and behavioural HVA. They do not mention that correlations between cortical and behavioural VMA were also explored, but that no correlation was found. This is made clear in the discussion, and potential explanations are offered. However, it is perhaps a little misleading to include the HVA findings in the abstract, without also reporting on the VMA. Especially when the abstract concludes: “These data reveal a tight link between cortical anatomy and visual perception at the level of individual observer and stimulus location.”, without a caveat.

– The reason we did not include it in the abstract is because we are limited to 150 words. In the abstract, we are explicit that our results are for the HVA alone. A lack of correlation for the VMA is not something we are trying to gloss over. We expand on this in our revised Discussion (**Lines 479-484**) and Conclusion (**Lines 516-518**).

– We have softened our abstract:

Line 26: ‘...these data reveal a ~~tight~~ link between cortical anatomy and visual perception’

2. [L264]: The authors state that “After factoring out between-observer variability, there was still a positive relation between local contrast sensitivity and local V1 surface area ($r_p = .78$, $x_{0.95} = .79$)”. Here the Spearman’s rho value is less than the 95th percentile of the bootstrapped distribution. The authors do not highlight this or justify why this result would nevertheless “indicate a robust effect of polar angle”.

– Please note we have revised this section (**Lines 218-271**) to include cut-offs from 2 bootstrap distributions. We find that the correlation in **Figure 4A** is above these two cut-offs, indicating a significant effect of both polar angle and observer on the correlation. We now visualize the unique contributions of polar angle and observer in **Figure 4B and C**.

3. [L429]: The discussion is rather limited. For example, the authors discuss the possible origin of various asymmetries. While the role of midget ganglion cells is mentioned, there’s no clear conclusion on the extent to which this determines the various reported asymmetries and individual differences in contrast sensitivity. I would prefer to see a more in-depth discussion of this.

– We have now expanded the discussion (**Lines 452-459**).

4. Another issue to address: In the present study, contrast sensitivity is measured only for one stimulus size and near vertically oriented gratings at one spatial frequency. Would the present results generalize to other orientations, spatial frequencies, and stimulus sizes? Is it possible that reported patterns would change for different types of stimuli, or stimuli in context rather than on a uniform gray background.

– We now discuss this (**Line 492-507**).

5. This touches upon the stimulus used to measure the pRFs and thus to estimate cortical magnification. This stimulus contained a variety of patterns “The pRF stimulus consisted of image patterns presented within a bar aperture that swept across the screen. The image patterns consisted of colorful objects, faces, and scenes at multiple scales that were superimposed on an achromatic pink noise (1/f)

background.” Please discuss whether this difference between psychophysical stimulus and pRF stimulus may affect results in any way. The surface area estimates relied on pRF estimates, which can be strongly affected by stimulus properties (e.g. DOI: 10.1146/annurev-vision-091517-033948 / DOI: 10.1016/j.neuroimage.2017.06.073).

– It is true pRF estimates can be affected by stimulus properties. However, this is not the case for polar angle measurements, which are the key measurement in defining our wedge-ROIs. Our recent work has shown that pRF polar angle estimates are highly reproducible across independent datasets that vary in many ways, with a near perfect vertex-wise correlation between two independent datasets (Himmelberg et al. 2021 *NeuroImage*).

– Further, the HVA and VMA for cortical magnification has been reproduced across four independent datasets: The NYU Retinotopy dataset (N=44), the HCP dataset (N=181), work from Silva et al., 2018, *NeuroImage* (N=22), and in an unpublished dataset (to be presented at VSS 2022) that we have analyzed in collaboration with Stanford University (N=24). These datasets differ in the stimulus carrier (image patterns described here, or black and white checkerboards), the maximum stimulus extent (8 deg, 12 deg, 7 deg), aperture (wedges, rings, and bars, or bars alone), temporal frequency, and fMRI protocol used to acquire the data. Thus, the asymmetries are robust to differences, and we think the correlation would remain with different pRF stimulus formats.

6. Consider whether it would not have been better to assess cortical surface using the same stimulus as used in the psychophysics (or alternatively, somehow assess contrast sensitivity for a range of patterns or stimuli and on an achromatic noise background).

– We do not think it would have made a difference for the same reason as above; pRF polar angle estimates are used to define the wedge-ROI, and polar angle measurements are highly robust across the stimulus carrier, aperture, and fMRI protocol. Further, in previous work, we have shown that the behavioral asymmetries are present for a broad variety of Gabor stimuli, including when presented with distractors.

7. [L44]: “Visual performance also changes as a function of polar angle” - the term visual performance is a little vague, the authors should consider making it clear what aspects of vision they are referring to or give examples (i.e., contrast sensitivity).

– Revised text referring to contrast sensitivity (**Line 43**).

8. [L99]: “We limit the range to 8° because the functional data are less reliable near the edge of the retinotopic mapping stimulus (12.4°)” - if this (8 deg) cutoff has been determined by previous studies, please provide a citation. If it is arbitrary or the parameters the authors have found to be useful, please state this.

– We now clarify this and provide the appropriate references (**Lines 106-107**).

9. [L113]: Figure 1B: It states that “The y-axes are matched so that the scaling in (B) is 4x the size of that in panel (A).” This should be 100x ($0.2 \text{ m}^2 = 200.000 \text{ mm}^2$).

– Corrected (**Line 122-123**).

10. [L114]: The (n=29) is positioned a bit in a confusing way, as it reads as it refers to hemispheres, whereas it concerns the # of observers.

– We have now amended the caption (**Line 122**).

11. [L113]: Figure 1D, left plot of polar angle. The two inflated cortices are placed on top of each other, and I find it difficult to distinguish where one surface starts and the other begins. It would be easier to compare the two if they did not overlap, perhaps by stacking them vertically. The same problem applies to the figure on the right of eccentricity.

– We have now adjusted **Figure 1D** so that the inflated surfaces are vertically stacked.

12. [L139]: Here the authors explain how the asymmetry index is calculated, but only using text. Mathematical equations would be easier to interpret and look-up. It would also be useful to explain what the index means: i.e., how would we interpret a VMA index of 0 compared to 100? Moreover, it seems very little is done with these indices. For example, the statistics reported next are done using the original contrast sensitivity values.

– Agreed. We moved these explanations down to the HVA and VMA correlations in **Figure 5** and have added equations and interpretation (**Lines 275-297**).

13. [L203]: In figure 3, cs is plotted on the x-axis, whereas surface is plotted on the y-axis. The reverse would make more sense, and would also more easily show the described non-linear relationship between surface area and cs. The inset figure is almost too tiny to be useful. Why not add a panel next to the figure to show it, if it is deemed important.

– We have expanded the inset as a panel as suggested (**Figure 3**). We would prefer to keep contrast sensitivity on the x-axis and the surface area on the y-axis to be consistent with all the other figures in the manuscript.

14. [L250]: Again, it is not clear from this paragraph (or the supplementary figures it refers to) what is being bootstrapped, and how the values in Figures S4A, S4B and S4C are different from each other and how they are calculated. I assume that for:

Figure S4A: raw CS and wedge-ROI-SA values are shuffled

Figure S4B: CS & wedge-ROI-SA values, whose between-observer variance has been removed are shuffled

Figure S4C: CS & wedge-ROI-SA values, whose within-observer variance has been removed are shuffled. This could all be made clearer.

– The reviewer's assumption is correct. However, we have now completed a new bootstrapping analysis and have ensured clarity in what values are being used and how they are shuffled (**Lines 227-244**).

15. [L378]: The authors highlight an interesting question raised by their findings: how variation in size of V1 relates to neural circuitry. They cite (47) stating “the size of V1 is inversely correlated with the size of its pRFs, suggesting that a larger V1 enables finer sampling of visual space”. Given the authors calculated pRF as part of the main analyses and raised this interesting point in the discussion, it is perhaps a shame that they did not also compare wedge-ROI-surface area with pRF size. This would help answer the question

they raise in this paragraph. A related general question is whether the size of their Gabors is optimal for each putative “V1 sampling size”.

– We agree that pRF size is an interesting property, which is why we raised it in the Discussion. However, for the Results, we felt it better to emphasize cortical magnification as a measure, rather than pRF size. There are several reasons. First, it has been hypothesized that the number of neurons (indexed by surface area, or cortical magnification) predicts contrast sensitivity (Virsu and Rovamo, 1979). It is not clear what predictions would be made from pRF size. One paper (Kay et al. 2015) found that larger pRF sizes were associated with better decoding precision of spatial position (when comparing across tasks within observers). Other work found that smaller pRFs are associated with better Vernier acuity when comparing between individuals (Song et al. 2015). Hence, the prediction is not straightforward and would require a computational model, which is beyond the scope of the current paper. A second issue is that cortical magnification depends on measuring pRF position and this has been shown to be more robust a measure than pRF size (Lerma-Usabiaga et al. 2020)

– Regarding the Gabor size; the stimuli used to measure pRFs were spatially broadband, and we did not attempt to match the stimuli used in psychophysics with the pRF mapping. We cannot be certain that we obtain the same results with different stimuli used for the psychophysics, however, we think it would be the case, based on our prior research showing that individual differences in contrast sensitivity tend to be robust to a variety of stimulus configurations, including stimulus size (Himmelberg et al. 2020).

16. [L411]: Here, the authors list evidence for many asymmetries in perception. However, there’s little to no attempt at discussing or integrating this information. E.g. Song et al. (2013), already cited by the authors, suggest certain trade-offs affecting surface area (spatial detail vs. contextual influences). Is there a tradeoff in which contrast sensitivity fits as well? I would like to read the author's educated guesses on this.

– These studies drew conclusions about trade-offs by testing different visual parameters. We tested one parameter and focused on testing different angle locations. Our study was not designed to look at any trade off. Contrast sensitivity is the currency of the visual system, and theoretically, greater contrast sensitivity should result in *better* performance on most tasks, rather than any trade-off. However, we know of one such trade off in behavior: the spatial properties of motion perception depend on contrast. The optimal size for perceiving motion *decreases* with *increasing* contrast (Tadin & Lapin, 2005, *Vision Research*); i.e., motion is better detected for larger patterns when contrast is low. This is because receptive field size decreases with stimulus contrast, and inhibitory surround mechanisms prevail over spatial summation. This trade off could be assessed in future work looking at pRF size, rather than cortical surface area.

17. [L681]: How exactly are wedge-ROIs calculated from retinotopic data? By selecting vertices whose pRF centers fall inside the wedges in the visual field, or only those vertices whose pRF is entirely inside the wedges in the visual field?

– The wedge-ROI calculation is detailed in our new Methods section (*Cortical magnification as a function of polar angle*) on **Lines 726-797**.

18. [L894]: the reference listed is to a biorxiv manuscript, but there's already an accepted publication associated with it.

– Now updated (**Line 1044**).

19. [L902]: Since the authors use the Psychophysics toolbox and the Eyelink, it is likely that they also use the code of the EyelinkToolbox. If so, acknowledge this by citing <https://doi.org/10.3758/BF03195489>

– Thanks, now updated with the citation (**Lines 551**).

REVIEWER COMMENTS

Reviewer #1 (Remarks to the Author):

The authors thoroughly addressed all my concerns.

However, I have an additional concern left, regarding the new analysis presented in Supplementary Figure 1: The median consistency of the behavioural HVA across blocks is .55. The behavioural-cortical correlation is slightly *higher* than that (.62 for pial).

Please briefly discuss why you do not consider this a 'Voodoo' correlation. It may be worth showing something like Spearman-Brown corrected predictions for the reliability of the full behavioural HVA data.

A minor comment is that 'equal number of trials' would be more precise wording in line 583.

Reviewer #2 (Remarks to the Author):

The authors have adequately addressed my comments and those of the other reviewers. I have not further comments. Nice work.

Reviewer #3 (Remarks to the Author):

I am mostly satisfied by the responses of the authors.
One remains though:

In their current point 5, I asked: Please discuss whether this difference between psychophysical stimulus and pRF stimulus may affect results in any way.

They discuss this, but only in the rebuttal, not the manuscript. In my view, it's sufficiently relevant to the manuscript to warrant inclusion.

An small additional remark concerns line Line 449-450, where the authors write:
we are currently assessing how cortical magnification changes as a function of polar angle in children.
[1]
[2] sEP:

I do not consider an announcement like this appropriate. I suggest to rephrase to: How cortical magnification changes as a function of polar angle in children still needs to be determined.

Response to Reviewers: Linking individual differences in human primary visual cortex to contrast sensitivity around the visual field

We would like to thank the reviewers for their constructive feedback on our revised manuscript. We have taken all their final comments into consideration. Changes to the manuscript have been tracked in blue text and are noted below. We hope that you now find that the manuscript is ready for publication.

Best,

Marc Himmelberg, Postdoctoral Research Associate

Jonathan Winawer, Associate Professor of Psychology and Neural Science

Marisa Carrasco, Julius Silver Professor of Psychology and Neural Science

Reviewer 1.

1. The authors thoroughly addressed all my concerns. However, I have an additional concern left, regarding the new analysis presented in Supplementary Figure 1: The median consistency of the behavioural HVA across blocks is .55. The behavioural-cortical correlation is slightly *higher* than that (.62 for pial). Please briefly discuss why you do not consider this a 'Voodoo' correlation. It may be worth showing something like Spearman-Brown corrected predictions for the reliability of the full behavioural HVA data.

The reviewer notes that the brain-behavior HVA correlation is higher than block-by-block behavioral HVA correlation, thus, it could be a voodoo correlation. We do not consider this a voodoo correlation for the following reasons. First, we are *not* selecting our cortical HVA based on how well it correlates with the behavioral HVA (like the voodoo studies argument). We are selecting it based on its spatial localisation to the psychophysical target. Second, the correlations across blocks are computed from approximately split-half data segments (40% vs 40%), whereas the brain data are correlated with the average from 100% of the behavioral data. As a result, the SNR of the behavioral data correlated with brain measures is considerably higher, and so a higher brain-behavior HVA correlation is not implausible. See Lage-Castellanos et al, 2019, eLife for an explicit argument about why split-half correlations can be lower than correlations with model-based predictions.

2. A minor comment is that 'equal number of trials' would be more precise wording in line 583.

We have amended the text on Line 652 to read 'equal number of trials'.

Reviewer 3.

1. In their current point 5, I asked: Please discuss whether this difference between psychophysical stimulus and pRF stimulus may affect results in any way. They discuss this, but only in the rebuttal, not the manuscript. In my view, it's sufficiently relevant to the manuscript to warrant inclusion.

We now discuss the points from our previous rebuttal in the manuscript (Lines 564-568).

Likewise, cortical polar angle asymmetries have been reproduced across several independent datasets that differ in their experimental design, including differences in the pRF stimulus carrier image (Benson et al. 2021; Himmelberg et al. 2021; Silva et al. 2018). The cortical asymmetries are robust to experimental differences because they rely on polar angle pRF measurements that have shown to be highly reproducible

across retinotopy experiments (Himmelberg et al. 2021). As these behavioral and cortical asymmetries are preserved across an array of stimulus conditions, we predict that the link between brain and behavioral measurements would also be preserved, albeit with modulations to the strength of the correlations.'

2. A small additional remark concerns line Line 449-450, where the authors write: we are currently assessing how cortical magnification changes as a function of polar angle in children. I do not consider an announcement like this appropriate. I suggest to rephrase to: How cortical magnification changes as a function of polar angle in children still needs to be determined.

We have amended the text on **Line 514** to read '*How cortical magnification changes as a function of polar angle in children still needs to be determined*'.